# Graph Fairness Learning under Distribution Shifts

## ABSTRACT

Graph neural networks (GNNs) have achieved remarkable performance on graph-structured data. However, GNNs may inherit prejudice from the training data and make discriminatory predictions based on sensitive attributes, such as gender and skin color. Recently, there has been an increasing interest in ensuring fairness on GNNs, while all of them are under the assumption that the training and testing data are under the same distribution. *Will graph fairness issue also emerge under distribution shifts? How does distribution shifts affect graph fairness learning?* All these open questions are largely unexplored from a theoretical perspective. In this work, we theoretically prove that graph fairness learning is determined by two key factors: the feature difference among certain groups and a fairness-related structure property of the graph. We further establish the relationship between fairness on the testing graph and two factors: fairness on the training graph, as well as the distribution difference between the training graph and the testing graph. Motivated by our theoretical analysis, we propose our framework FatraGNN to ensure fairness performance under distribution shifts on graphs. Specifically, we use a graph generator to generate graphs that result in prediction unfairness and are under different distributions. Then we maximize the alignment of representations between the training graph and generated graphs for each certain group. This empowers us to attain high classification and fairness performance even on generated graphs exhibiting significant unfairness, thereby enhancing the ability to handle the testing graphs effectively. Experiments on real-world and semi-synthetic datasets validate that our model can achieve better classification and fairness performance compared with other state-of-the-art baselines.

## KEYWORDS

Graph Neural Networks, Fairness, Distribution Shifts

**ACM Reference Format:**
Anonymous Author(s). 2018. Graph Fairness Learning under Distribution Shifts. In *Proceedings of Make sure to enter the correct conference title from your rights confirmation emai (Conference acronym 'XX).* ACM, New York, NY, USA, 16 pages. https://doi.org/XXXXXXX.XXXXXXX

## 1 INTRODUCTION

Graph Neural Networks (GNNs) are powerful deep learning algorithms that can be used to model graph-structured data. In recent years, there have been enormous successful applications of GNNs on various areas such as social media mining [18], drug discovery

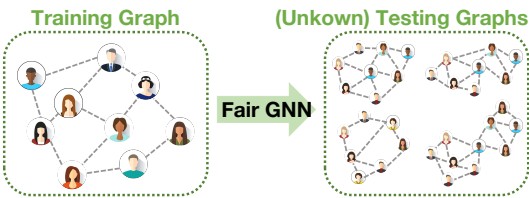

**Figure 1: Example of fairness under distribution shifts on graphs.**

[21], and recommender system [5, 47]. However, despite their success, there is a growing concern that GNNs may inherit or even amplify discrimination and social bias from the training data, leading to unfair treatment of certain groups with sensitive attributes such as gender, age, region, and skin color. This phenomenon may lead to social and ethical issues, thus limiting the application of GNNs in critical areas such as job marketing [22], criminal justice [38], and credit scoring [16].

To mitigate the unfairness issue of GNNs, many fair GNNs have been proposed. For example, [45, 48] add an extra fairness-related term to the optimization objective to improve fairness performance of the model. [6, 10] adopt adversarial learning to learn node representations that are not distinguishable on sensitive attributes by fooling the discriminator. [12, 28] aim to achieve fairness by feeding GNNs with less biased graphs. Despite the success of fair GNNs, they are all proposed under the common hypothesis that the training and testing data are under the identical distribution, which does not always hold in reality.

Actually, the distribution shifts scenario is very common in the real world [2, 4, 26, 44]. For example, as shown in Figure 1, a fair job recommendation system trained on a social network drawn from one state may be utilized on social networks drawn from other states. However, the structures and features of social networks in different states may be under different distributions, so the fair GNN may not adapt to the new network, leading to discrimination such as disproportionately recommending low-payment jobs to certain sensitive groups identified by race or gender. Previous studies [14, 44] mainly aim at keeping the stable classification performance of GNNs under distribution shifts, while largely ignoring the fairness issue. *Why graph fairness issue emerges under distribution shifts? How distribution shifts affect the fairness of GNNs?* The answers from a theoretical and methodological perspective remain largely unknown.

Recently, the fairness learning problem under distribution shifts has received considerable attention [3, 20, 36]. However, all these works focus on Euclidean data, and cannot utilize the crucial structure information of graph-structured data for prediction. In this work, we first theoretically analyze the relationship between graph data distribution and graph fairness (Theorem 3.6), and conclude that graph fairness is determined by sensitive structure property and the feature difference between different sensitive groups. This explains why the graph unfairness issue happens under distribution

shifts. Then we prove that there are two factors affecting the graph fairness on the testing graph: one is the fairness on the training graph, and the other is the representation distances between the training graph and the testing graph of each certain group (Theorem 3.8). These findings well deepen our understanding of the fairness learning of graphs under distribution shifts.

Motivated by our theoretical insights, we further propose a novel model called FatraGNN to handle the problem. Our model employs an adversarial module to ensure fairness on the training graph. To ensure fairness on the unknown testing graphs, we first utilize a graph generation module to generate graphs that are prone to causing significant unfairness and are under different distributions, and then maximize the alignment of the representations between the training graph and the generated graphs of each certain group. Based on Theorem 3.8, if our model can learn fair representations for these generated graphs with large biases, it will be more robust to distribution shifts and perform better on specific testing graphs which usually have smaller biases. In summary, our contributions are three-fold:

- To the best of our knowledge, this is the first attempt to study fairness learning on graphs under distribution shifts from a theoretical perspective. We theoretically analyze the relationship between graph fairness and graph data distribution and discover the key factors that affect fairness learning under distribution shifts.
- Based on the theoretical insights, we propose our FatraGNN, which consists of an adversarial debiasing module, a graph generation module, and an alignment module, to ensure fairness on the unknown testing graphs.
- Extensive experiments show that our FatraGNN outperforms state-of-the-art baselines under distribution shifts in terms of classification and fairness performance on real-world and semi-synthetic datasets.

## 2 PRELIMINARIES AND NOTATIONS

Let $\mathcal{G} = (\mathcal{V}, \mathcal{E}, \mathbf{X})$ be a graph with $n$ nodes, where $\mathcal{V} = \{v_1, v_2, \ldots, v_n\}$ is the node set, $\mathcal{E} \subseteq \mathcal{V} \times \mathcal{V}$ is the edge set. $\mathbf{X} = [x_1, x_2, \ldots, x_n] \in \mathbb{R}^{n \times \zeta}$ represents the node feature matrix, where $x_i$ is the feature vector of node $v_i$ and $\zeta$ is the dimension of node features. Graph structure of $\mathcal{G}$ can be described by the adjacency matrix $\mathbf{A} \in \mathbb{R}^{n \times n}$, and $\mathbf{A}_{ij} = 1$ iff there exists an edge between nodes $v_i$ and $v_j$. The diagonal degree matrix is denoted as $\mathbf{D} = \text{diag}(d_1, \cdots, d_n)$, where $d_i = \sum_j \mathbf{A}_{ij}$. Node sensitive attributes are specified by the $t$-th channel of $\mathbf{X}$, i.e., $\mathbf{F} = \mathbf{X}_{:,t} = [f_1, f_2, \ldots, f_n] \in \{0, 1\}^n$, where $f_i$ is the sensitive attribute of node $i$. Here we focus on binary classification tasks, and the binary labels of the nodes are denoted by $\mathbf{Y} \in \{0, 1\}^n$.

**Fairness Metric** There exist several different definitions of fairness, such as group fairness [30], individual fairness [13], counterfactual fairness [25], and degree-related fairness [34]. In this work, we focus on group fairness, which can be measured by two metrics: equalized oods [19] and demographic parity [7]. Equalized odds seeks to achieve the same true positive rate and true negative rate between two sensitive groups. It is defined as $\Delta_{EO} = \frac{1}{2} \sum_{y=0}^{1} |\mathbb{E}_{v_i \in \mathcal{V}}(\hat{y}_i = y | y_i = y, f_i = 1) - \mathbb{E}_{v_j \in \mathcal{V}}(\hat{y}_j = y | y_j = y, f_j = 0)|$, where $\hat{y}_i$ is the predicted label of $v_i$. Demographic parity measures the acceptance rate difference between two sensitive groups.

For example, in binary classification tasks such as deciding whether a student should be admitted into a university or not, demographic parity is considered to be achieved if the model yields the same acceptance rate for individuals in both sensitive groups. It is defined as $\Delta_{DP} = |\mathbb{E}_{v_i \in \mathcal{V}}(\hat{y}_i = 1 | f_i = 1) - \mathbb{E}_{v_j \in \mathcal{V}}(\hat{y}_j = 1 | f_j = 0)|$.

**Fairness on Graph Distribution Shifts** Following the definition of previous study [44], we characterize the data generation process as $\mathbb{P}(\mathbf{A}, \mathbf{X}, \mathbf{Y}|\mathbf{e}) = \mathbb{P}(\mathbf{A}, \mathbf{X}|\mathbf{e})\mathbb{P}(\mathbf{Y}|\mathbf{A}, \mathbf{X}, \mathbf{e})$, where $\mathbf{e}$ represents a random variable denoting the latent environmental factors that influence the data distribution. First, the graph is generated via $\mathbb{P}(\mathbf{A}, \mathbf{X}|\mathbf{e})$. Then the labels are generated via $\mathbb{P}(\mathbf{Y}|\mathbf{A}, \mathbf{X}, \mathbf{e})$. We assume that $\mathbb{P}(\mathbf{Y}|\mathbf{A}, \mathbf{X}, \mathbf{e})$ is invariant under different environments. Our aim is to achieve a scenario where the generation of $\mathbf{Y}$ is not influenced by the features related to sensitive attributes $\mathbf{F}$ to ensure fairness. We consider training graph from data distribution $\mathbb{P}(\mathbf{A}_{\mathcal{S}}, \mathbf{X}_{\mathcal{S}}|\mathbf{e} = \mathcal{S})$, testing graphs from data distribution $\mathbb{P}(\mathbf{A}_{\mathcal{T}}, \mathbf{X}_{\mathcal{T}}|\mathbf{e} = \mathcal{T})$. This work intends to ensure fairness when $\mathcal{S} \neq \mathcal{T}$.

## 3 GRAPH FAIRNESS UNDER DISTRIBUTION SHIFTS

In this section, we first establish a relationship between graph fairness and graph data distribution $\mathbb{P}(\mathbf{A}, \mathbf{X}|\mathbf{e})$. Then we gain insight into why distribution shifts may lead to fairness degradation.

### 3.1 Relationship between Data Distribution and Graph Fairness

We first use aggregation feature distance to establish the connection between data distribution and graph fairness. Referring to commonly used GNNs, we define the aggregation features as $\mathbf{H} = \tilde{\mathbf{D}}^{-1}\tilde{\mathbf{A}}\mathbf{X}$, where $\tilde{\mathbf{D}} = \mathbf{D} + \mathbf{I}$, $\tilde{\mathbf{A}} = \mathbf{A} + \mathbf{I}$. For the convenience of expression, we define sensitive group, EO group, and aggregation feature distance between EO groups as follows:

*Definition 3.1.* (Sensitive group) The sensitive group of nodes with sensitive attribute $f$ is defined as:

$$\mathcal{V}_f = \{v_i \in \mathcal{V}|f_i = f\}. \tag{1}$$

*Definition 3.2.* (EO group) The EO group of nodes with label $y$ and sensitive attribute $f$ is defined as:

$$\mathcal{V}_f^y = \{v_i \in \mathcal{V}|(f_i = f) \cup (y_i = y)\}. \tag{2}$$

*Definition 3.3.* (Aggregation feature distance between sensitive groups with the same label) The aggregation feature distance between sensitive groups with label $y$ is defined as:

$$\eta_y = \max_{v_a \in \mathcal{V}_1^y} \min_{v_b \in \mathcal{V}_0^y} ||h_a - h_b||_2, \tag{3}$$

where $h_a$ and $h_b$ are the aggregation features of nodes $v_a$ and $v_b$, respectively.

The aggregation feature distance $\eta_y$ defines the maximum shortest path from a node in $\mathcal{V}_1^y$ to a node in $\mathcal{V}_0^y$, thus can measure the aggregation feature difference between the two sensitive groups with label $y$. Large $\eta_y$ implies that the aggregation features of different sensitive groups are easy to distinguish, and GNNs may make predictions based on this sensitive information, resulting in unfairness.

We then show that $\eta_y$ is mainly affected by two factors determined by $\mathbb{P}(\mathbf{A}, \mathbf{X}|\mathbf{e})$. The first factor is the sensitive structure property of the graph. Previous fairness studies [28, 43] focus on the sensitive homophily of the graph structure, defined as $\alpha = \mathbb{E}_{v_i \in \mathcal{V}} \frac{\sum_{j \in N_i \cup \{v_i\}} \mathbf{1}_{(f_i=f_j)}}{d_i + 1}$, where $N_i$ is the neighbors of node $v_i$, $\mathbf{1}_{(f_i=f_j)}$ is the indicator function evaluating to 1 if and only if $f_i = f_j$. They believe that higher sensitive homophily will make the aggregation features of two sensitive groups more distinguishable, resulting in unfairness. However, we find that lower sensitive homophily will also make aggregation features of sensitive groups distinguishable. For example, the graph in Figure 2 has very low sensitive homophily according to $\alpha$. After the aggregation step of GNN, different sensitive groups may change their features but are still distinguishable. We further point out that $\eta_y$ is determined by whether the nodes tend to have balanced neighborhoods, i.e., the number of neighbors belonging to different sensitive groups is nearly the same. To theoretically analyze the relationship between balanced neighborhoods and $\eta_y$, we define a new sensitive balance degree to quantify the structure property:

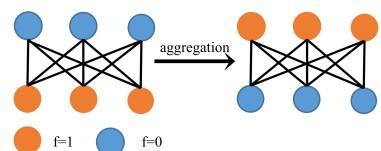



f=1   f=0



**Figure 2: Example of low sensitive homophily.**

*Definition 3.4.* (Sensitive balance degree) The sensitive balance degree of node $v_i$ with sensitive attribute $f_i$ is:

$$u_i = |p_i - q_i|, \tag{4}$$

where $p_i = \frac{\sum_{j \in N_i \cup \{v_i\}} \mathbf{1}_{(f_i=f_j)}}{d_i + 1}$ and $q_i = \frac{\sum_{j \in N_i \cup \{v_i\}} \mathbf{1}_{(f_i \neq f_j)}}{d_i + 1}$ represent the proportions of neighbors with the same and different sensitive attribute, respectively. The average sensitive balance degree on a graph is :

$$u = \mathbb{E}_{i \in \mathcal{V}}(u_i). \tag{5}$$

The sensitive balance degree reflects the difference in the number of neighbors around node $v_i$ belonging to different sensitive groups. If a node has nearly the same number of neighbors with different sensitive attributes, then it has a more balanced neighborhood and smaller $u_i$, and vice versa.

The second factor that affects $\eta_y$ is the feature difference between different sensitive groups. We assume features of nodes belonging to two sensitive groups follow Gaussian distribution, i.e., $\mathbb{P}(x_a \mid v_a \in \mathcal{V}_1) \sim \mathcal{N}(\mu_{\mathcal{V}_1} \mathbf{I}_\zeta, \sigma_{\mathcal{V}_1}^2 \mathbf{I}_\zeta)$ and $\mathbb{P}(x_b \mid v_b \in \mathcal{V}_0) \sim \mathcal{N}(\mu_{\mathcal{V}_0} \mathbf{I}_\zeta, \sigma_{\mathcal{V}_0}^2 \mathbf{I}_\zeta)$. With the feature distribution of two sensitive groups and the graph structure property $u$, we can bound $\eta_y$ with the following theorem:

THEOREM 3.5. *For any $\delta \in (0, 1)$, with probability greater than $1 - \delta$ and large enough feature dimension $\zeta$, we have:*

$$\eta_y^2 \geq (\sigma_{\mathcal{V}_1}^2 + \sigma_{\mathcal{V}_0}^2)\zeta(1 - 2\sqrt{\frac{log(2/\delta)}{\zeta}}) + \zeta u^2(\mu_{\mathcal{V}_1} - \mu_{\mathcal{V}_0})^2,$$
$$\eta_y^2 \leq (\sigma_{\mathcal{V}_1}^2 + \sigma_{\mathcal{V}_0}^2)\zeta(1 + 4\sqrt{\frac{log(2/\delta)}{\zeta}}) + \zeta u^2(\mu_{\mathcal{V}_1} - \mu_{\mathcal{V}_0})^2. \tag{6}$$

From the above theorem, we can find out that the upper bound and lower bound of $\eta^y$ are determined by $(\mu_{\mathcal{V}_1} - \mu_{\mathcal{V}_0})$ and $u$. Then we show that the fairness metric $\Delta_{EO}$ is actually bounded by $\eta_y$ as follows.

THEOREM 3.6. *Consider an encoder $g : H \to Z \in \mathbb{R}^{n \times \zeta'}$ extracting $\zeta'$-dimensional representations $Z$ and an classifier $\omega : Z \to C \in \mathbb{R}^{n \times 2}$ predicting the binary labels of the nodes. Assume that $g$ and $\omega$ have $L_1$-Lipschitz and $L_2$-Lipschitz continuity, respectively, then equalized odds is bounded by:*

$$\Delta_{EO} \leq L_1 L_2 \frac{\sum_{y=0}^1 \eta_y}{2}. \tag{7}$$

Combining Theorem 3.6 and Theorem 3.5, we find out that $\Delta_{EO}$ is mainly affected by two key factors determined by $\mathbb{P}(\mathbf{A}, \mathbf{X}|\mathbf{e})$: 1) The difference of features between sensitive groups $\mu_{\mathcal{V}_1} - \mu_{\mathcal{V}_0}$. Larger $\mu_{\mathcal{V}_1} - \mu_{\mathcal{V}_0}$ indicates that the features of the two sensitive groups are easier to distinguish, resulting in larger $\Delta_{EO}$. 2) The average sensitive balance degree of the graph $u$. Larger $u$ implies the nodes in the graph tend to have unbalanced neighbors, resulting in larger $\Delta_{EO}$.

We direct the readers to Appendix A for proofs of all the above theorems.

## 3.2 Bounds on Fairness on the Testing Graph

Given the factors that affect graph fairness, we can gain insight into the reason why distribution shifts may lead to unfairness.

As $\Delta_{EO}$ is determined by two factors affected by $\mathbb{P}(\mathbf{A}, \mathbf{X}|\mathbf{e})$, suppose the data distribution differs between the training graph and testing graphs, i.e., $\mathbb{P}(\mathbf{A}_\mathcal{S}, \mathbf{X}_\mathcal{S}|\mathbf{e} = \mathcal{S}) \neq \mathbb{P}_\mathcal{T}(\mathbf{A}_\mathcal{T}, \mathbf{X}_\mathcal{T}|\mathbf{e} = \mathcal{T})$, then the two factors including $u$ and $(\mu_{\mathcal{V}_1} - \mu_{\mathcal{V}_0})$ will also change, resulting in fairness deterioration in some cases. For example, if $(\mu_{\mathcal{V}_1} - \mu_{\mathcal{V}_0})$ and $u$ are small in the training graph but large in the testing graph, then the model is highly fair with lower $\Delta_{EO}$ in the training graph but highly unfair with larger $\Delta_{EO}$ in the testing graph.

Then we characterize the difference of $\Delta_{EO}$ between the training graph and the testing graph, denoted as $\Delta_{EO}^\mathcal{S} - \Delta_{EO}^\mathcal{T}$, by analyzing the accuracy difference between the training graph and the testing graph of each EO group. The EO group in the testing graph with label $y$ and sensitive attribute $f$ is denoted as $\mathcal{T}_f^y = \{v_i \in \mathcal{V}_\mathcal{T}|(f_i = f) \cap (y_i = y)\}$, where $\mathcal{V}_\mathcal{T}$ is the node set in the testing grah, and we define the prediction accuracy on $\mathcal{T}_f^y$ as $\mathbb{E}_{\mathcal{T}_f^y} = \mathbb{E}_{v_i \in \mathcal{T}_f^y}(\hat{y}_i = y)$. Similarly, on training graph we have $\mathbb{E}_{\mathcal{S}_f^y} = \mathbb{E}_{v_i \in \mathcal{S}_f^y}(\hat{y}_i = y)$. Then we bound the equalized odds difference for data in the training graph and testing graph as:

$$\Delta_{EO}^\mathcal{T} - \Delta_{EO}^\mathcal{S} \leq \sum_{y,f} |\mathbb{E}_{\mathcal{T}_f^y} - \mathbb{E}_{\mathcal{S}_f^y}|. \tag{8}$$

We then define EO group representation distance between the training graph and the testing graph in Definition 3.7, and build a relationship between the representation distance and $|\mathbb{E}_{\mathcal{T}_f^y} - \mathbb{E}_{\mathcal{S}_f^y}|$ in Theorem 3.8.

*Definition 3.7.* (EO group representation distance between the training graph and the testing graph) For EO group with label $y$ and sensitive attribute $f$, we define the representation distance between the training graph and the testing graph as:

$$\epsilon_f^y = \max_{v_j \in \mathcal{T}_f^y} \min_{v_i \in \mathcal{S}_f^y} ||z_i - z_j||_2, \tag{9}$$

where $z_i$ is the representation of node $v_i$ learned by the encoder $g$.

THEOREM 3.8. *Assume that the nonlinear transformation $\omega(Z) = RELU(ZW_\omega)$ has $L_2$-Lipschitz continuity, we have:*

$$|\mathbb{E}_{\mathcal{T}_f^y} - \mathbb{E}_{\mathcal{S}_f^y}| \le L_2 \epsilon_f^y. \tag{10}$$

*Then equalized odds difference between the training graph and the testing graph can be bounded as:*

$$\Delta_{EO}^{\mathcal{T}} - \Delta_{EO}^{\mathcal{S}} \le L_2 \sum_{f,y} \epsilon_f^y. \tag{11}$$

Based on Theorem 3.8, we can see that $\Delta_{EO}^{\mathcal{T}}$ relies on both $\Delta_{EO}^{\mathcal{S}}$ and EO group representation distance, which is determined by how much $\mathbb{P}(\mathbf{A}_{\mathcal{T}}, \mathbf{X}_{\mathcal{T}}|e = \mathcal{T})$ differs from $\mathbb{P}(\mathbf{A}_{\mathcal{S}}, \mathbf{X}_{\mathcal{S}}|e = \mathcal{S})$.

To alleviate the unfairness issue in the testing graph, i.e., minimize $\Delta_{EO}^{\mathcal{T}}$, we not only have to minimize $\Delta_{EO}^{\mathcal{S}}$, but also have to minimize the EO group representation distance. Also, minimizing the EO group representation distance leads to the minimization of $\Delta_{DP}^{\mathcal{T}}$. Detailed proofs of minimization of $\Delta_{DP}^{\mathcal{T}}$, Theorem 3.8, and Eq. (8) are deferred to Appendix A.

## 4 METHODOLOGY

In order to solve the unfairness under distribution shifts problem, motivated by the findings in Section 3, we present our framework (shown in Figure 3), which mainly includes three parts: (a) the generative adversarial debiasing module to get smaller $\Delta_{EO}^{\mathcal{S}}$ in the training graph, (b) the graph generation module to generate graphs that lead to unfairness and are under different distributions, (c) the EO group alignment module to minimize the EO group representation distance.

### 4.1 Adversarial Debiasing on Training Graph

As suggested in Theorem 3.8, to improve fairness on the testing graph, we have to ensure fairness on the training graph. Combining the aggregation step and the encoder $g$ discussed in Section 3.1, we use a GNN-based encoder $\kappa_{\theta_\kappa} : (\mathbf{A}, \mathbf{X}) \to Z \in \mathbb{R}^{n \times \zeta'}$ with parameters $\theta_\kappa$ to extract the $\zeta'$-dimensional representations of the nodes. If the representations of different sensitive groups are distinguishable, then the classifier may make predictions based on this information, resulting in unfairness. In order to make the representations undistinguishable, we use a sensitive discriminator $\xi_{\theta_\xi} : Z \to F \in \{0, 1\}^n$ with parameters $\theta_\xi$ to predict the sensitive attributes of the nodes given their representations $Z$. And the encoder $\kappa_{\theta_\kappa}$ is trained to learn similar representations between sensitive groups, thus can fool the discriminator. Leveraging adversarial training, we compute the loss of the discriminator and encoder as:

$$\min_{\theta_\kappa} \max_{\theta_\xi} \mathcal{L}_\xi = \mathbb{E}_{v_i \in \mathcal{S}} (f_i \log(\xi_{\theta_\xi}(\kappa_{\theta_\kappa}(x_i)) + (1 - f_i) \log(1 - \xi_{\theta_\xi}(\kappa_{\theta_\kappa}(x_i)))). \tag{12}$$

Besides, we also train the encoder together with an MLP-based classifier $\omega_{\theta_\omega}$ to minimize the classification loss to ensure accuracy:

$$\min_{\theta_\kappa, \theta_\omega} \mathcal{L}_c = -\mathbb{E}_{v_i \in \mathcal{S}} (y_i \log(\hat{y}_i) + (1 - y_i) \log(1 - \hat{y}_i)). \tag{13}$$

### 4.2 Graph Generation Module

To address the unfairness issue caused by data distribution shifts, we should also get similar representations between the training graph and the testing graph for each EO group, as suggested in Theorem 3.8. During the training process, we consider extending the training graph to graphs under the distribution of $\mathbb{P}(\mathbf{A}_{\mathcal{T}}, \mathbf{X}_{\mathcal{T}}|e = \mathcal{T})$ and training $\kappa_{\theta_\kappa}$ to learn similar representations for graphs under the distribution of $\mathbb{P}(\mathbf{A}_{\mathcal{T}}, \mathbf{X}_{\mathcal{T}}|e = \mathcal{T})$ and $\mathbb{P}(\mathbf{A}_{\mathcal{S}}, \mathbf{X}_{\mathcal{S}}|e = \mathcal{S})$. As $\mathbb{P}(\mathbf{A}_{\mathcal{T}}, \mathbf{X}_{\mathcal{T}}|e = \mathcal{T})$ is unknown during the training process, it is challenging to generate the exact testing graphs, so we generate the graphs which result in unfairness and are under different distributions. If our model can handle graphs that are more likely to cause unfairness, it will be able to address fairness issues under distribution shifts more effectively. The generated graphs follows distribution $\mathbb{P}(\mathbf{A}_{\mathcal{T}'}, \mathbf{X}_{\mathcal{T}'}|e = \mathcal{T}')$. As demonstrated in Theorem 3.6, larger $\mu_{\mathcal{V}_1} - \mu_{\mathcal{V}_0}$ and $u$ will lead to poor fairness performance. So we propose a graph generation module, including a structure modification step to generate $\mathbf{A}_{\mathcal{T}'}$ by modifying $\mathbf{A}_{\mathcal{S}}$, and a feature generator to generate $\mathbf{X}_{\mathcal{T}'}$ based on $\mathbf{X}_{\mathcal{S}}$. Thus we can generate graphs that will lead to unfairness and are under different distributions.

For the structure modification step, two strategies can be employed. One is to randomly add edges between nodes with the same sensitive attribute and remove edges between nodes with different sensitive attribute. The other is to randomly add edges between nodes with different sensitive attribute and remove edges between nodes with the same sensitive attribute. Both are used to get a bunch of generated $\mathbf{A}_{\mathcal{T}'}$ with larger $u$ before training, resulting in unbalanced neighborhoods.

To make our model adapt to various structures, we feed one of the generated graphs into training during every certain number of epochs, then we use an MLP-based feature generator $\Gamma_{\theta_\Gamma} : \mathbf{X}_{\mathcal{S}} \to \mathbf{X}_{\mathcal{T}'}$ to generate features. The generated graph with $\mathbf{A}_{\mathcal{T}'}$ and $\mathbf{X}_{\mathcal{T}'}$ is then feed into $\kappa_{\theta_\kappa}$ and $\omega_{\theta_\omega}$ to make predictions. We also include a regularization term to ensure that the feature generator does not produce features that significantly stray from the features of the training graph. The feature generator is trained to maximize the fairness loss:

$$\max_{\theta_\Gamma} \mathcal{L}_{fair}^{\mathcal{T}'} = \frac{1}{2} \sum_{y=0}^{1} |\mathbb{E}_{v_i \in \mathcal{S}_1^y}(\hat{y}_i = y|\Gamma(x_i), \mathbf{A}_{\mathcal{T}'}) \tag{14}$$
$$-\mathbb{E}_{v_j \in \mathcal{S}_0^y}(\hat{y}_j = y|\Gamma(x_j), \mathbf{A}_{\mathcal{T}'})| - \tau ||\mathbf{X}_{\mathcal{T}'} - \mathbf{X}_{\mathcal{S}}||_F^2.$$

where $|| \cdot ||_F^2$ is the Frobenius norm of matrix, $\tau$ is the coefficient.

Thus the feature generator can be trained to explore the features that lead to poor fairness performance but not deviate too much from the training graph. After the structure modification step and the feature generation step, we can generate graphs which lead to unfairness and are under different distributions.

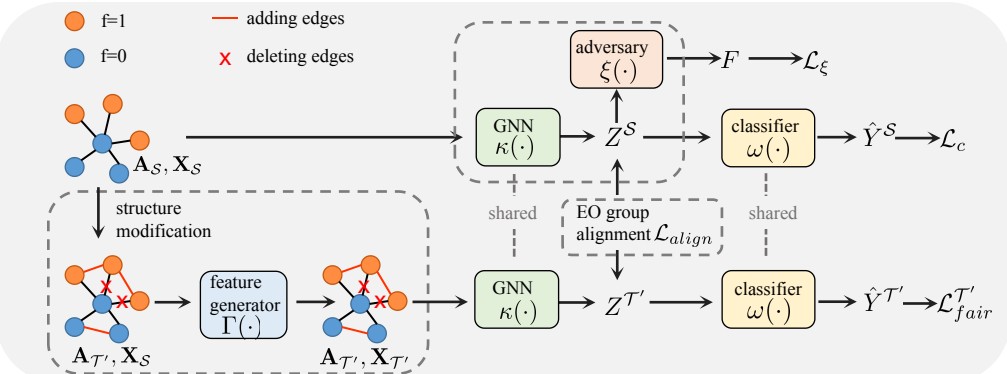

**Figure 3: An overview of FatraGNN.**

## 4.3 EO Group Alignment Module

We can learn from Theorem 3.8 that unfairness issue in the testing graph can be alleviated by minimizing the EO group representation distance $\epsilon_f^y$. Thus we should minimize EO group representation distance between the training graph and the generated graph.

We utilize a similarity score $\lambda_f^y = \mathbb{E}_{i \in \mathcal{S}_f^y, j \in \mathcal{T'}_f^y} \frac{(z_i z_j)}{||z_i|| \cdot ||z_j||}$ to measure the alignment of EO group representation. Higher $\lambda_f^y$ implies better alignment and lower $\epsilon_f^y$. Then we maximize the similarity score of all EO groups:

$$\max_{\theta_\kappa} \mathcal{L}_{align} = \sum_{f,y} \lambda_f^y. \tag{15}$$

In this way, we can get smaller $\epsilon_f^y$, implying better fairness performance in the testing graph. Furthermore, the alignment module ensures the accuracy of classification on generated graphs by guiding the GNN-based encoder to acquire similar representations for both the training graph and the generated graphs. Consequently, this enhancement leads to improved classification accuracy on the testing graphs.

Meanwhile, the alignment module implicitly aids in preserving the causal features from disruption. The alignment module forces the encoder to learn similar representations for the generated graphs and the training graph, and the encoder for the generated graphs and the training graph is shared, so it is not possible to learn similar representations for features with significant differences. This implicitly forces the generator to learn features that are not significantly different from the original graph but may cause unfairness. Experimental analysis can be found in Section 5.4.

## 5 EXPERIMENT

**Datasets** We use five datasets to evaluate the performance of our model under distribution shifts. Each dataset comprises at least two graphs: one for training and validation, and others for testing. The five datasets comprise three real-world datasets and two semi-synthetic datasets summarized as follows. 1) **Pokecs** includes Pokec-z and Pokec-n, which are drawn from the popular social network in Slovakia [11] based on the provinces that users belong

to. Both Pokec-z and Pokec-n consist of users belonging to two major regions of the corresponding provinces. We use Pokec-z for training and validation, and Pokec-n for testing. We treat "region" as the sensitive attribute, and the task is to predict the working field of the users. 2) **Bail-Bs** is obtained from the commonly used fairness-related graph Bail [23], where nodes are defendants released on bail. Utilizing the modularity-based community detection method [31], we partition Bail into communities and find that they exhibit different data distributions. Then we retain five large communities and name them from B0 to B4. We use B0 for training and validation, and the remaining graphs B1 to B4 for testing. The task is to decide whether to bail the defendants with "race" being the sensitive attribute. 3) **Credit-Cs** is partitioned the same way as Bail-Bs from Credit [46], where nodes represent credit card users. We get five communities named from C0 to C4. C0 is used for training and validation, while C1 to C4 are used for testing. The task is to classify the credit risk of the clients as high or low with "age" being the sensitive attribute. 4) **sync-B1s** comprises testing graphs with different $u$ from 0 to 0.6 obtained by modifying the structure of B1, and a training graph B0. 5) **sync-B2s** consisits of testing graphs obtained by modifying B2 the same way as sync-B1s and a training graph B0. More details such as dataset statistics can be found in Appendix B.

**Baselines** We compare our model with six baselines: 1) Traditional learning methods: MLP [32], GCN [24]. 2) Fair GNNs: FairVGNN [43], NIFTY [1], EDITS [12]. 3) Out-of-distribution (OOD) GNN: EERM [44]. 4) Model-agnostic OOD methods: SWAD [8], SAGM [42].

**Performance Evaluation** We use accuracy (ACC) and ROC-AUC to evaluate the predictive performance of the node classification task. To measure fairness, we use $\Delta_{DP}$ and $\Delta_{EO}$ introduced in Section 2. Note that a model with lower $\Delta_{DP}$ and $\Delta_{EO}$ implies better fairness performance. To comprehensively assess the classification and fairness performance of a model across various testing graphs, we introduce a metric denoted as $s = \text{ACC} + \text{ROC-AUC} - \Delta_{DP} - \Delta_{EO}$, where greater values of this metric indicate superior model performance. We calculate the total score for each method by summing up their scores on all testing graphs, and then provide the overall rankings for each method.

**Table 1: Classification and fairness performance (%±$\sigma$) on Bail-Bs. ↑ denotes the larger, the better; ↓ denotes the opposite. Best ones are in bold.**

| | metric | MLP | GCN | FairVGNN | NIFTY | EDITS | EERM | SWAD | SAGM | FatraGNN (ours) |
|---|---|---|---|---|---|---|---|---|---|---|
| B1 | ACC↑ | 70.53±1.01 | 72.93±4.06 | 69.76±2.03 | 69.54±7.26 | 72.69±1.72 | 73.25±1.4 | 70.45±3.78 | 73.08±4.25 | **74.59±0.93** |
| | ROC-AUC↑ | 62.76±1.87 | 59.41±14.42 | 64.82±4.32 | 62.65±5.95 | 59.91±0.31 | 63.98±1.28 | 62.33±2.54 | 62.76±3.45 | **66.0±0.01** |
| | $\Delta_{DP}$ ↓ | 4.83±9.38 | 4.58±0.78 | 11.05±4.58 | 7.21±4.54 | 4.35±1.3 | 8.85±2.57 | 8.33±5.43 | 7.33±4.59 | **1.14±2.87** |
| | $\Delta_{EO}$ ↓ | 7.48±7.31 | 10.19±2.3 | 8.35±4.82 | 9.57±2.8 | 9.22±0.97 | 10.93±2.38 | 6.29±3.98 | 7.35±4.56 | **2.38±3.19** |
| B2 | ACC↑ | 64.33±0.63 | 69.88±0.45 | 65.03±2.4 | 69.95±8.3 | 69.03±0.16 | 70.2±0.12 | 68.35±5.63 | 68.67±3.24 | **70.46±0.44** |
| | ROC-AUC↑ | 59.21±1.18 | 68.35±10.68 | 70.21±2.61 | 65.93±13.46 | 74.25±0.73 | 72.23±0.49 | 66.82±4.86 | 70.67±2.14 | **73.27±4.48** |
| | $\Delta_{DP}$ ↓ | 8.36±1.62 | 6.91±0.58 | 5.64±2.78 | 3.21±4.54 | 3.2±3.06 | 8.31±0.5 | 6.34±5.22 | 5.78±2.53 | **0.15±0.79** |
| | $\Delta_{EO}$ ↓ | 6.51±0.32 | 8.68±0.2 | 3.23±3.47 | 3.57±2.8 | 2.89±0.54 | 6.29±0.12 | 7.48±6.73 | 6.34±3.56 | **0.43±1.14** |
| B3 | ACC↑ | 60.76±0.18 | 68.56±4.2 | 70.63±0.61 | 68.8±9.76 | 68.56±1.82 | 70.69±5.42 | 70.45±4.86 | 69.50±2.12 | **71.65±4.65** |
| | ROC-AUC↑ | 62.89±2.87 | 72.99±0.68 | 80.76±5.01 | 77.98±5.5 | 79.28±1.48 | 79.98±3.61 | 78.64±5.67 | 78.43±3.90 | **82.17±3.63** |
| | $\Delta_{DP}$ ↓ | 9.8±0.38 | 12.72±2.44 | 8.05±0.45 | 6.21±4.54 | 5.24±0.03 | 5.64±3.49 | 7.21±2.36 | 6.78±3.23 | **5.02±3.54** |
| | $\Delta_{EO}$ ↓ | 6.29±0.36 | 14.15±3.09 | 9.18±0.36 | 5.57±2.8 | 3.08±0.27 | 4.65±1.21 | 4.37±5.35 | 5.67±2.84 | **2.43±4.94** |
| B4 | ACC↑ | 63.13±1.69 | 69.43±0.48 | 68.99±2.44 | 57.96±11.99 | 68.42±0.14 | 70.9±1.36 | 70.03±9.87 | 70.88±0.98 | **72.59±3.39** |
| | ROC-AUC↑ | 61.57±0.97 | 76.4±0.78 | 77.23±1.14 | 69.21±5.39 | 69.2±1.41 | 68.81±2.27 | 68.90±4.56 | 69.34±1.89 | **77.36±3.79** |
| | $\Delta_{DP}$ ↓ | 4.45±3.15 | 4.49±1.13 | 5.21±6.03 | 3.21±4.54 | 3.2±9.1 | 7.23±0.26 | 5.34±4.26 | 6.36±6.32 | **2.48±3.09** |
| | $\Delta_{EO}$ ↓ | 3.29±3.54 | 8.74±1.62 | 5.33±6.18 | 2.57±2.8 | 5.6±7.86 | 9.04±0.86 | 4.37±3.54 | 7.34±4.67 | **2.45±6.67** |
| | rank | 9 | 8 | 3 | 7 | 2 | 5 | 6 | 4 | **1** |

**Table 2: Quantitative results (%±$\sigma$) on Pokecs. (bold: best)**

| | metric | MLP | GCN | FairVGNN | NIFTY | SWAD | SAGM | FatraGNN (ours) |
|---|---|---|---|---|---|---|---|---|
| Pokec-n | ACC↑ | 52.74±3.67 | 54.83±2.34 | 60.8±0.54 | 58.68±5.54 | 59.25±0.89 | 58.78±2.33 | **62.00±0.24** |
| | ROC-AUC↑ | 65.38±0.43 | 63.48±2.34 | 65.26±1.45 | 67.09±2.25 | 66.37±1.34 | 65.67±2.45 | **67.82±3.23** |
| | $\Delta_{DP}$ ↓ | 4.86±1.23 | 7.38±0.28 | 5.88±2.34 | 4.21±3.43 | 7.45±3.87 | 5.67±3.22 | **1.34±0.27** |
| | $\Delta_{EO}$ ↓ | 4.16±2.34 | 6.37±0.52 | 6.26±2.21 | 3.82±3.88 | 6.36±4.56 | 4.19±2.45 | **1.43±2.68** |
| | rank | 6 | 7 | 4 | 2 | 5 | 3 | **1** |

**Experimental Setting** We perform a hyperparameter search for our model on all dataset groups. For other baseline models: GCN, MLP, FairVGNN, NIFTY, EDITS, and EERM, we carefully fine-tune them to get optimal performance on all the dataset groups. Note that EDITS and EERM have higher complexity and are hard to be trained on Pokec-z , so we only report the results of other baselines on Pokecs. For all methods, we randomly run 5 times and report the mean and variance of each metric. More details such as the hyperparameter setting can be found in Appendix B.

## 5.1 Evaluation on Real-world Datasets

We use three real-world datasets for evaluation: Pokecs, Bail-Bs, and Credit-Cs.

**Results** Table 1 and Table 2 show the effectiveness of FatraGNN in terms of classification and fairness performance on all testing graphs in Bail-Bs and Pokecs. Due to space limitations, we defer the results on Credit-Cs to Appendix C.

We observe that the proposed FatraGNN outperforms all baselines in most cases. Additionally, we find that while fairness baselines aim to improve fairness performance, they cannot perform well on testing graphs when distribution shifts. Although the graph OOD model EERM achieves better classification performance than fairness baselines when distribution shifts, it has lower fairness performance on all the testing graphs because it cannot learn fair representations.

We also analyze the relationship between accuracy and $\Delta_{EO}$ of the models, because good fairness performance could be a result of poor classification performance. For example, if a model mis-classifies all samples, then the accuracy on all EO groups will be

0, resulting in $\Delta_{EO} = 0$, which implies good fairness performance. However, this is not the ideal model. Fairness models may ensure fairness at the cost of accuracy, so we further show the Pareto front curves [37], which are generated by a grid search of hyperparameters, to show this trade-off between accuracy and $\Delta_{EO}$. As shown in Figure 4, the horizontal axis represents $\Delta_{EO}$ and the vertical axis represents accuracy. Curves closer to the upper-left corner imply higher accuracy and lower $\Delta_{EO}$, indicating better trade-off performance. We can see that FatraGNN achieves better performance than fairness baselines in terms of this trade-off.

## 5.2 Evaluation on Semi-synthetic Datasets

We further use sync-B1s and sync-B2s to test the performance of each method on testing graphs with different $u$. Testing graphs with higher $u$ have less balanced neighbors and may result in unfairness.

**Results** As $u$ is calculated by $|p - q|$, we find that accuracy of the models have different changing trend when $p - q > 0$ and $p - q < 0$. In order to demonstrate the performance of models more clearly, we use $u' = p - q$ instead to reflect the average sensitive balance degree of the graphs.

The classification and fairness performance are shown in Figure 5. Overall, our FatraGNN outperforms other baselines in terms of both accuracy and $\Delta_{EO}$ on most testing graphs. Moreover, FatraGNN demonstrates low variance in both classification and fairness performance across different testing graphs with various $u'$, indicating its potential to perform well when distribution shifts. Additionally, we find that most models achieve their optimal fairness performance when $u'$ is close to 0. When $u = |u'|$ increases, $\Delta_{EO}$ also increases, verifying our analysis in Section 3.1 that unbalanced neighborhoods will lead to unfairness. Additionally, we find that baselines tend to

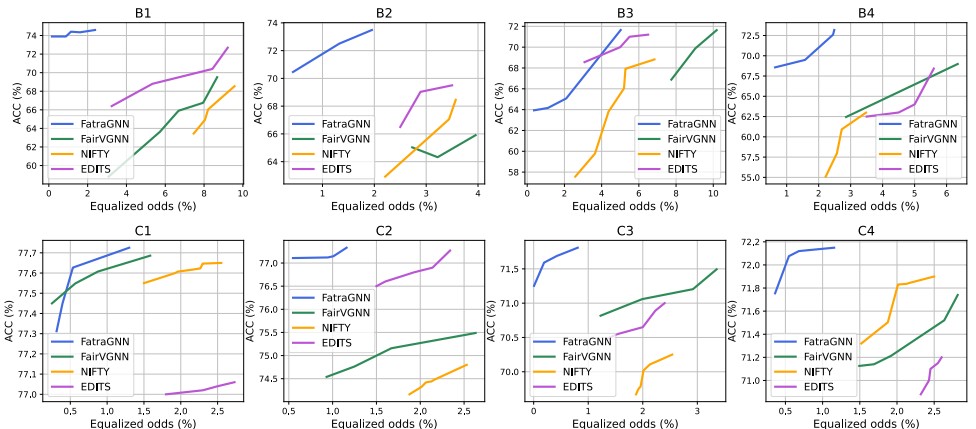

**Figure 4: Trade-off of ACC and $\Delta_{EO}$ on all testing graphs of Bail-Bs, Credit-Cs. Upper-left corner (high accuracy, low $\Delta_{EO}$) is preferred. The first row shows the results on B1 to B4. The second row shows the results on C1 to C4.**

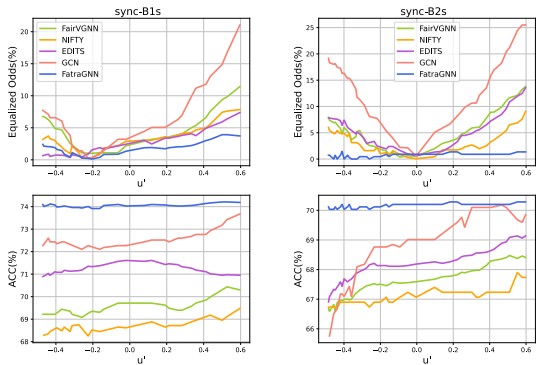

**Figure 5: Accuracy and $\Delta_{EO}$ on sync-B1s (left) and sync-B2s (right).**

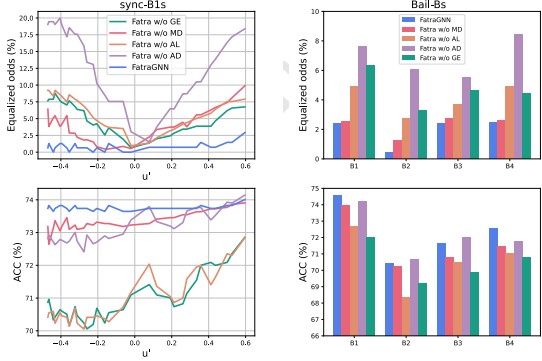

**Figure 6: Ablation study on sync-B1s (left) and Bail-Bs (right). Please note that testing graphs of sync-B1s have continuously varying $u'$, so we utilize a line chart to illustrate the performance change of the model on graphs with varying $u'$.**

achieve better accuracy as $u'$ increases. This is because the nodes with the same sensitive attribute tend to share the same label, which provides additional information for the classification task.

## 5.3 Ablation Study

To fully understand the effect of each component of FatraGNN on alleviating unfairness under distribution shifts, we propose several variants of FatraGNN, including **Fatra w/o AD** as removing the adversarial module, **Fatra w/o GE** as removing the graph generation module, **Fatra w/o MD** as removing the structure modification step, and **Fatra w/o AL** as removing the EO group alignment module. Results of the ablation study on sync-B1s and Bail-Bs are shown in Figure 6. We can see that FatraGNN consistently outperforms the other variants. Without the adversarial module, Fatra w/o AD learns distinguishable representations for the two sensitive groups, resulting in poor fairness performance. Without the graph generation module, Fatra w/o GE fails to perform well when distribution shifts. Without the alignment module, Fatra w/o AL only generates graphs but is not trained to learn similar representations between the input graph and the generated graph for each EO group, resulting in similar poor performance as Fatra w/o GE. Without modification of the structure, Fatra w/o MD still performs better than Fatra w/o GE and Fatra w/o AL, since it is trained to adapt to different feature distributions. However, due to the lack of training graph with different $u'$, Fatra w/o MD cannot perform well on testing graphs with different $u'$.

## 5.4 Additional Analysises

**Analysis of Generated Graphs** We also analyze the generated graphs and find that the generated graphs have different distributions from the training graph, and their casual features are maintained.

First, we show that the generated graphs are under different distributions from the training graph. Usually, graphs with different $u$ and $\mu_{\mathcal{V}_1} - \mu_{\mathcal{V}_0}$ have different distributions because the structures of the graphs and the feature difference between different sensitive groups are different. In the generation module, we modify the structure of the graph to get larger $u$. Also, as we learn feature representations through an end-to-end method to get graphs that lead to unfairness, $\mu_{\mathcal{V}_1} - \mu_{\mathcal{V}_0}$ of the generated graph will also change during training. We do experiments on Credit-Cs, Bail-Bs and Pokecs. As shown in Figure 7, as the number of epochs increases, $\mu_{\mathcal{V}_1} - \mu_{\mathcal{V}_0}$

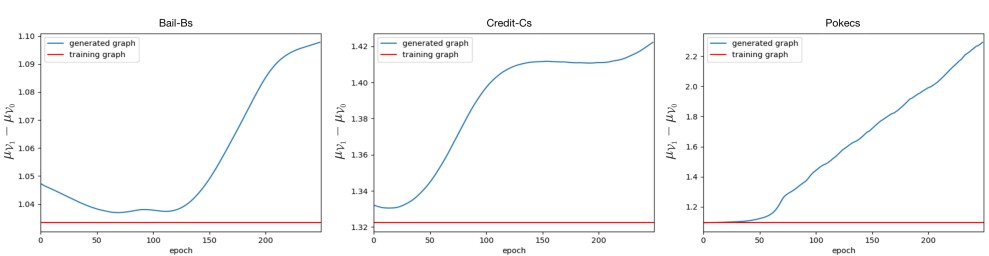

Figure 7: Change of $\mu_{\mathcal{V}_1} - \mu_{\mathcal{V}_0}$ on the generated graphs during training on Bail-Bs, Credit-Cs, and Pokecs.

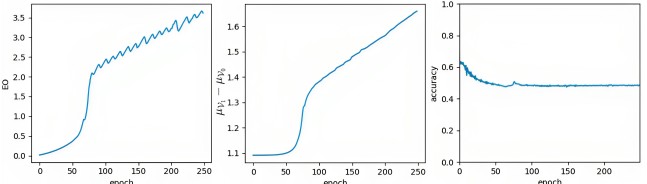

Figure 8: The representation difference between sensitive groups, equalized odds, and accuracy on the generated graph during training.

increases, indicating that the generated graphs are under different distributions from the training graph.

Then we examine if there are potential disruptions in the causal features of the generated graphs. We conduct experiments on Bail-Bs to observe the changes in $\mu_{\mathcal{V}_1} - \mu_{\mathcal{V}_0}$, $\Delta EO$, and accuracy during the training process. As shown in Figure 8, during training, the generated graphs have larger $\mu_{\mathcal{V}_1} - \mu_{\mathcal{V}_0}$ and poorer fairness (larger $\Delta EO$). This suggests that the generated graphs will lead to unfairness and are under different distributions. Still, the accuracy hardly decreases, indicating that the key features of the graph are not disrupted.

**Analysis of Representation Distances** To demonstrate that by achieving alignment of representations between the training graph and generated graphs, our model can ensure alignment between the representations of the training graph and testing graphs for better fairness and accuracy performance, we plot the representations of the training graph and testing graphs of Bail-Bs using t-SNE [39]. The implementation details and result figures can be found in Appendix C.4. We can see that the representations of nodes belonging to the same EO group on the training graph and testing graphs are close, indicating that by minimizing the representation difference between the training graph and the generated graphs, the alignment module of our model can ensure the proximity of representations of the same EO group between the training graph and testing graphs, thereby guaranteeing fairness and accuracy on the testing graphs.

**Analysis of Convergence** We notice that during training, it is not difficult to tune the parameters to achieve convergence. Despite that no theory can guarantee convergence to the saddle point, it functions well in our experiments, which has also been observed in many other adversarial methods [40, 41, 43]. Additional experiments are provided in Appendix C.3.

Other additional experiment results such as hyperparameter study can be found in Appendix C.

## 6 RELATED WORK

**Fairness on GNNs** There have been a number of works focused on the unfairness problem on graphs [1, 12, 28, 43]. NIFTY [1] conducts a two-level strategy to modify GNN to ensure fairness and stability. EDITS [12] proposes to debias the attributed network to achieve fairness by feeding GNNs with less biased graphs. FairVGNN [43] proposes a framework to improve fairness by automatically identifying and masking sensitive-correlated features considering correlation variation after each feature propagation step. [27] indicates that fairness on a graph is contingent on both the size of the sensitive groups and the connected situation of a graph, and proposes FairAdj to learn a fair matrix to achieve dynamic fairness and prediction utility. More recently, Graphair [28] adopts adversarial learning and contrastive learning to automatically discover fairness-aware augmentations from input graphs. However, these works are all under the assumption that training and testing data are under the same distribution.

**Fairness under Distribution Shifts** Recently, a number of works intend to study fairness under distribution shifts [3, 9, 15, 29, 35, 36]. [9, 17] reweight the examples in the training data to approximate the proportions of groups in the testing data. [29, 35] consider data in the testing data as combinations of samples in the training data with arbitrary weights and ensure fairness of the model under the worst-case shift. [3] derives a sufficient condition for transferring fairness, and proposes a self-training algorithm to minimize and balance consistency loss across groups. However, these works are all focused on Euclidean data and ignore the special property of graph structure. To the best of our knowledge, this is the first work that considers fairness under distribution shifts on graphs.

More related work such as OOD generalization methods on graphs can be found in Appendix D.

## 7 CONCLUSION

In this work, we study the unfairness problem under distribution shifts on graphs, which is crucial for the real-world applications of fair GNNs. We theoretically prove that graph fairness is determined by a sensitive structure property and feature difference between sensitive groups of the graph, and explain the reason why distribution shifts will lead to unfairness. We then derive an upper bound for fairness on the testing graph. Based on our analysis, we further propose a novel FatraGNN framework to alleviate this problem. Experimental results demonstrate that FatraGNN consistently outperforms state-of-the-art baselines in terms of fairness-accuracy trade-off performance under distribution shifts.

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

# A PROOFS

## A.1 Proof of Theorem 3.5

**Theorem 3.5.** *For any $\delta \in (0, 1)$, with probability greater than $1 - \delta$ and large enough feature dimension $\zeta$, we have:*

$$\eta_y^2 \geq (\sigma_{\mathcal{V}_1}^2 + \sigma_{\mathcal{V}_0}^2)\zeta(1 - 2\sqrt{\frac{\log(2/\delta)}{\zeta}}) + \zeta u^2(\mu_{\mathcal{V}_1} - \mu_{\mathcal{V}_0})^2,$$

$$\eta_y^2 \leq (\sigma_{\mathcal{V}_1}^2 + \sigma_{\mathcal{V}_0}^2)\zeta(1 + 4\sqrt{\frac{\log(2/\delta)}{\zeta}}) + \zeta u^2(\mu_{\mathcal{V}_1} - \mu_{\mathcal{V}_0})^2. \tag{16}$$

*Proof.*

To get the upper bound and lower bound of $\eta_y$, we introduce a function $\eta(a, b)$ describing the aggregation feature difference between any two nodes belonging to $\mathcal{V}_1$ and $\mathcal{V}_0$, i.e., $\eta(a, b) = ||h_a - h_b||_2$, where $h_a$ and $h_b$ are aggregation features of node $v_a \in \mathcal{V}_1$ and $v_b \in \mathcal{V}_0$, respectively. So $\eta_y = \max_{v_a \in \mathcal{V}_1^y} \min_{v_b \in \mathcal{V}_0^y} ||h_a - h_b||_2$ equals $\eta(a, b)$ with specific $a$ and $b$. Thus we can bound $\eta_y$ by bounding $\eta(a, b)$. As $\eta(a, b) = ||h_a - h_b||_2$, we first give the distribution of $(h_a - h_b)$.

Recall that we assume $\mathbb{P}(x_a \mid v_a \in \mathcal{V}_1) \sim \mathcal{N}(\mu_{\mathcal{V}_1}\mathbf{I}_\zeta, \sigma_{\mathcal{V}_1}^2\mathbf{I}_\zeta)$ and $\mathbb{P}(x_b \mid v_b \in \mathcal{V}_0) \sim \mathcal{N}(\mu_{\mathcal{V}_0}\mathbf{I}_\zeta, \sigma_{\mathcal{V}_0}^2\mathbf{I}_\zeta)$. As $h_a = \frac{1}{d_a+1}\sum_{j \in N_a \cup \{v_a\}} x_j$, $h_b = \frac{1}{d_b+1}\sum_{j \in N_b \cup \{v_b\}} x_j$, $h_a$ and $h_b$ follow the Gaussian distribution:

$$h_a \sim \mathcal{N}(p_a\mu_{\mathcal{V}_1} + q_a\mu_{\mathcal{V}_0}, (p_a\sigma_{\mathcal{V}_1}^2 + q_a\sigma_{\mathcal{V}_0}^2)\mathbf{I}_\zeta),$$

$$h_b \sim \mathcal{N}(p_b\mu_{\mathcal{V}_0} + q_b\mu_{\mathcal{V}_1}, (p_b\sigma_{\mathcal{V}_0}^2 + q_b\sigma_{\mathcal{V}_1}^2)\mathbf{I}_\zeta), \tag{17}$$

where $p_a = \sum_{v_j \in N_a \cup \{v_a\}} \frac{\mathbf{1}_{(f_a = f_j)}}{d_a+1}$, $q_a = \sum_{v_j \in N_a \cup \{v_a\}} \frac{\mathbf{1}_{(f_a \neq f_j)}}{d_a+1}$.

Then we have:

$$h_a - h_b \sim \mathcal{N}((p-q)(\mu_{\mathcal{V}_1} - \mu_{\mathcal{V}_0}), (\sigma_{\mathcal{V}_1}^2 + \sigma_{\mathcal{V}_0}^2)\mathbf{I}_\zeta). \tag{18}$$

where $p = \mathbb{E}_{i \in \mathcal{V}}(p_i)$, $q = \mathbb{E}_{i \in \mathcal{V}}(q_i)$. For the convenience of proof, we transform the above distribution into the form of a standard Gaussian distribution $[h_a - h_b - (p-q)(\mu_{F_1} - \mu_{F_0})] \sim \mathcal{N}(0, (\sigma_{F_1}^2 + \sigma_{F_0}^2)\mathbf{I}_\zeta)$. Then we give the $l_2$ norm of this standard distribution and subsequently analyze the range of it by Lemma A.1 and Corollary A.2.

$$||h_a - h_b - (p-q)(\mu_{\mathcal{V}_1} - \mu_{\mathcal{V}_0})||_2^2$$
$$= ||h_a - h_b||_2^2 - 2(p-q)(\mu_{\mathcal{V}_1} - \mu_{\mathcal{V}_0})$$
$$\sum_j (h_a[j] - h_b[j]) + \zeta(p-q)^2(\mu_{\mathcal{V}_1} - \mu_{\mathcal{V}_0})^2$$
$$= ||h_a - h_b||_2^2 - (p-q)(\mu_{\mathcal{V}_1} - \mu_{\mathcal{V}_0})$$
$$\sum_j \left(2(h_a[j] - h_b[j]) - (p-q)(\mu_{\mathcal{V}_1} - \mu_{\mathcal{V}_0})\right)$$
$$\approx ||h_a - h_b||_2^2 - \zeta(p-q)^2(\mu_{\mathcal{V}_1} - \mu_{\mathcal{V}_0})^2,$$

where $h_a[j]$ is the $j$-th dimension of $h_a$. We then state the following results on standard Gaussian distribution.

**Lemma A.1.** *(Laurent-Massart $\chi^2$ tail bound) Consider a standard Gaussian vector $\mathbf{z} \sim \mathcal{N}(0, \mathbf{I}_\zeta)$. For any positive vector $\mathbf{a} \in \mathbb{R}_{\leq 0}^\zeta$, and any $t \leq 0$, the following concentration holds.*

$$\mathbb{P}\left[\sum_{i=1}^\zeta \mathbf{a}_i \mathbf{z}_i^2 \geq ||\mathbf{a}||_1 + 2||\mathbf{a}||_2\sqrt{t} + 2||\mathbf{a}||_\infty t)\right] \leq \exp(-t),$$
$$\mathbb{P}\left[\sum_{i=1}^\zeta \mathbf{a}_i \mathbf{z}_i^2 \leq ||\mathbf{a}||_1 - 2||\mathbf{a}||_2\sqrt{t}\right] \leq \exp(-t). \tag{19}$$

The following corollary immediately follows from using $t = \log(2/\delta)$ and $\mathbf{a}_i = 1$ in the above lemma.

**Corollary A.2.** *($\ell_2$ norm of Gaussian vector). Consider $\mathbf{z} \sim \mathcal{N}(0, \sigma^2\mathbf{I}_\zeta)$, for any $\delta \in (0, 1)$ and large enough $\xi$, with probability greater than $1 - \delta$, we have:*

$$\sigma^2\zeta\left(1 - 2\sqrt{\frac{\log(2/\delta)}{\zeta}}\right) \leq ||\mathbf{z}||_2^2 \leq \sigma^2\zeta\left(1 + 4\sqrt{\frac{\log(2/\delta)}{\zeta}}\right). \tag{20}$$

As $[h_a - h_b - (p-q)(\mu_{F_1} - \mu_{F_0})] \sim \mathcal{N}(0, (\sigma_{F_1}^2 + \sigma_{F_0}^2)\mathbf{I}_\zeta)$, with probability greater than $1 - \delta$ and large enough $\zeta$, we have:

$$(\sigma_{\mathcal{V}_1}^2 + \sigma_{\mathcal{V}_0}^2)\zeta(1 - 2\sqrt{\frac{\log(2/\delta)}{\zeta}}) + \zeta(p-q)^2(\mu_{\mathcal{V}_1} - \mu_{\mathcal{V}_0})^2 \leq ||h_a - h_b||_2^2$$
$$\leq (\sigma_{\mathcal{V}_1}^2 + \sigma_{\mathcal{V}_0}^2)\zeta(1 + 4\sqrt{\frac{\log(2/\delta)}{\zeta}}) + \zeta(p-q)^2(\mu_{\mathcal{V}_1} - \mu_{\mathcal{V}_0})^2. \tag{21}$$

With $u = p - q$, we have:

$$\eta_y^2 \geq (\sigma_{\mathcal{V}_1}^2 + \sigma_{\mathcal{V}_0}^2)\zeta(1 - 2\sqrt{\frac{\log(2/\delta)}{\zeta}}) + \zeta u^2(\mu_{\mathcal{V}_1} - \mu_{\mathcal{V}_0})^2,$$

$$\eta_y^2 \leq (\sigma_{\mathcal{V}_1}^2 + \sigma_{\mathcal{V}_0}^2)\zeta(1 + 4\sqrt{\frac{\log(2/\delta)}{\zeta}}) + \zeta u^2(\mu_{\mathcal{V}_1} - \mu_{\mathcal{V}_0})^2. \tag{22}$$

This concludes the proof of the theorem.

## A.2 Proof of Eq. (8)

$$\Delta_{EO}^{\mathcal{T}} - \Delta_{EO}^{\mathcal{S}}$$
$$= (|\mathbb{E}_{\mathcal{T}_0^1} - \mathbb{E}_{\mathcal{T}_1^1}| + |\mathbb{E}_{\mathcal{T}_0^0} - \mathbb{E}_{\mathcal{T}_1^0}|) - (|\mathbb{E}_{\mathcal{S}_0^1} - \mathbb{E}_{\mathcal{S}_1^1}| + |\mathbb{E}_{\mathcal{S}_0^0} - \mathbb{E}_{\mathcal{S}_1^0}|)$$
$$= (|\mathbb{E}_{\mathcal{T}_0^1} - \mathbb{E}_{\mathcal{T}_1^1}| - |\mathbb{E}_{\mathcal{S}_0^1} - \mathbb{E}_{\mathcal{S}_1^1}|) + (|\mathbb{E}_{\mathcal{T}_0^0} - \mathbb{E}_{\mathcal{T}_1^0}| - |\mathbb{E}_{\mathcal{S}_0^0} - \mathbb{E}_{\mathcal{S}_1^0}|)$$
$$\overset{(a)}{\leq} \left||\mathbb{E}_{\mathcal{T}_0^1} - \mathbb{E}_{\mathcal{T}_1^1}| - |\mathbb{E}_{\mathcal{S}_0^1} - \mathbb{E}_{\mathcal{S}_1^1}|\right| + \left||\mathbb{E}_{\mathcal{T}_0^0} - \mathbb{E}_{\mathcal{T}_1^0}| - |\mathbb{E}_{\mathcal{S}_0^0} - \mathbb{E}_{\mathcal{S}_1^0}|\right|$$
$$\overset{(b)}{\leq} \sum_{y,f} |\mathbb{E}_{\mathcal{T}_f^y} - \mathbb{E}_{\mathcal{S}_f^y}|, \tag{23}$$

where inequality (a) and (b) hold due to $a - b \leq |a - b|$ and $||a - b| - |a' - b'|| \leq |a - a'| + |b - b'|$.

## A.3 Proof of Theorem 3.6

**Theorem 3.6.** *Consider an encoder $g : H \rightarrow Z \in \mathbb{R}^{n \times \zeta'}$ extracting $\zeta'$-dimensional representations $Z$ and a classifier $\omega : Z \rightarrow C \in \mathbb{R}^{n \times 2}$ predicting the binary labels of the nodes. Assume that $g$ and $\omega$ have $L_1$-Lipschitz and $L_2$-Lipschitz continuity, respectively, then equalized odds is bounded by:*

$$\Delta_{EO} \leq L_1 L_2 \frac{\sum_{y=0}^1 \eta_y}{2}. \tag{24}$$

*Proof.*

To characterize the relationship between $\mathcal{V}_1^y$ and $\mathcal{V}_0^y$, we utilize a partition that can separate the nodes in $\mathcal{V}_0^y$ into different sets $B_y^{(i)}$. Every node $v_j$ in $B_y^{(i)}$ is near to a certain node $v_i \in \mathcal{V}_1^y$, satisfying $||h_i - h_j||_2 \le \eta_y$. Obviously, $\mathcal{V}_0^y = \cup_{v_i \in \mathcal{V}_1^y} B_y^{(i)}$. We have

$$
\begin{aligned}
\Delta_{EO} &= \frac{1}{2} \sum_{y=0}^{1} |\mathbb{E}_{i \in \mathcal{V}_1^y}(\hat{y}_i = y_i) - \mathbb{E}_{j \in \mathcal{V}_0^y}(\hat{y}_j = y_j)| \\
&= \frac{1}{2} \sum_{y=0}^{1} |\mathbb{E}_{i \in \mathcal{V}_1^y} \omega \circ g(h_i)[y] - \mathbb{E}_{j \in \mathcal{V}_0^y} \omega \circ g(h_j)[y]| \\
&= \frac{1}{2} \sum_{y=0}^{1} \frac{1}{n_{\mathcal{V}_1^y}} \sum_{i \in \mathcal{V}_1^y} |\omega \circ g(h_i)[y] - \frac{1}{n_{B_y^{(i)}}} \sum_{j \in B_y^{(i)}} \omega \circ g(h_j)[y]| \\
&= \frac{1}{2} \sum_{y=0}^{1} \frac{1}{n_{\mathcal{V}_1^y}} \sum_{i \in \mathcal{V}_1^y} \frac{1}{n_{B_y^{(i)}}} \sum_{j \in B_y^{(i)}} |\omega \circ g(h_i)[y] - \omega \circ g(h_j)[y]|,
\end{aligned}
\tag{25}
$$

where $\omega \circ g(h_i)[y]$ is the $y$-th row of the vector $\omega \circ g(h_i)$.

As the nonliear transformation $g$ and classifier $\omega$ have $L_1$-Lipschitz and $L_2$-Lipschitz continuity, then $|\omega \circ g(h_i)[y] - \omega \circ g(h_j)[y]| = ||\omega \circ g(h_i) - \omega \circ g(h_j)||_\infty = \frac{1}{\sqrt{2}}||\omega \circ g(h_i) - \omega \circ g(h_j)||_2 \le \frac{L_2}{\sqrt{2}}||g(h_i) - g(h_j)||_2 \le \frac{L_1 L_2}{\sqrt{2}}||h_i - h_j|| \le \frac{L_1 L_2}{\sqrt{2}}\eta_y$. So $\Delta_{EO} \le L_1 L_2 \frac{\sum_{y=0}^{1} \eta_y}{2\sqrt{2}} \le L_1 L_2 \frac{\sum_{y=0}^{1} \eta_y}{2}$.

## A.4 Proof of Theorem 3.8

**Theorem 3.8.** Assume that the nonlinear transformation $\omega(Z) = RELU(ZW_\omega)$ has $L_2$-Lipschitz continuity, we have:

$$
|\mathbb{E}_{\mathcal{T}_f^y} - \mathbb{E}_{\mathcal{S}_f^y}| \le L_2 \epsilon_f^y.
\tag{26}
$$

Then equalized odds difference between the training graph and the testing graph can be bounded as:

$$
\Delta_{EO}^{\mathcal{T}} - \Delta_{EO}^{\mathcal{S}} \le L_2 \sum_{f,y} \epsilon_f^y.
\tag{27}
$$

*Proof.*

To characterize the relationship between $\mathcal{T}_f^y$ and $\mathcal{S}_f^y$, we utilize a partition that can separate the nodes in $\mathcal{T}_f^y$ into different sets $R_{f,y}^{(i)}$. Every node $v_j$ in $R_{f,y}^{(i)}$ is near to a certain node $v_i \in \mathcal{S}_f^y$, satisfying $||z_i - z_j||_2 \le \epsilon_f^y$. Obviously, $\mathcal{T}_f^y = \cup_{v_i \in \mathcal{S}_f^y} R_{f,y}^{(i)}$. As $\mathbb{E}_{\mathcal{S}_f^y} = \mathbb{E}_{v_i \in \mathcal{S}_f^y}(\omega(z_i)[y])$, $\mathbb{E}_{\mathcal{T}_f^y} = \mathbb{E}_{v_i \in \mathcal{T}_f^y}(\omega(z_i)[y])$, we have:

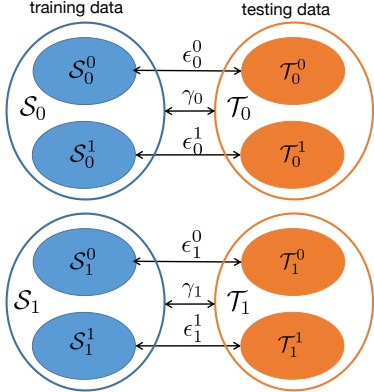

**Figure 9: Illustration of minimization of $\Delta_{DP}^{\mathcal{T}}$.**

$$
\begin{aligned}
|\mathbb{E}_{\mathcal{T}_f^y} - \mathbb{E}_{\mathcal{S}_f^y}| &= |\mathbb{E}_{i \in \mathcal{S}_f^y}(\hat{y}_i = y) - \mathbb{E}_{j \in \mathcal{T}_f^y}(\hat{y}_j = y)| \\
&= |\mathbb{E}_{i \in \mathcal{S}_f^y}(\omega(z_i)[y]) - \mathbb{E}_{j \in \mathcal{T}_f^y}(\omega(z_j)[y])| \\
&= \frac{1}{n_{\mathcal{S}_f^y}} \sum_{i \in \mathcal{S}_f^y} |\omega(z_i)[y] - \frac{1}{n_{R_{f,y}^{(i)}}} \sum_{j \in R_{f,y}^{(i)}} \omega(z_j)[y]| \\
&= \frac{1}{n_{\mathcal{S}_f^y}} \sum_{i \in \mathcal{S}_f^y} \frac{1}{n_{R_{f,y}^{(i)}}} \sum_{j \in R_{f,y}^{(i)}} |\omega(z_i)[y] - \omega(z_j)[y]|.
\end{aligned}
\tag{28}
$$

As $\omega$ has $L_2$-Lipschitz continuity, $|\omega(z_i)[y] - \omega(z_j)[y]| = ||\omega(z_i) - \omega(z_j)||_\infty = \frac{1}{\sqrt{2}}||\omega(z_i) - \omega(z_j)||_2 \le \frac{1}{\sqrt{2}}L_2||z_i - z_j|| \le \frac{1}{\sqrt{2}}L_2 \epsilon_f^y$. Then

$$
|\mathbb{E}_{\mathcal{T}_f^y} - \mathbb{E}_{\mathcal{S}_f^y}| \le \frac{1}{\sqrt{2}}L_2 \epsilon_f^y \le L_2 \epsilon_f^y.
\tag{29}
$$

According to Eq. (8) and the above equation, we have:

$$
\Delta_{EO}^{\mathcal{T}} - \Delta_{EO}^{\mathcal{S}} \le L_2 \sum_{f,y} \epsilon_f^y.
\tag{30}
$$

## A.5 Minimization of $\Delta_{DP}^{\mathcal{T}}$

Similar to Theorem 3.8 which indicates that $\Delta_{EO}^{\mathcal{T}} - \Delta_{EO}^{\mathcal{S}} \le L_2 \sum_{f,y} \epsilon_f^y$, we can also prove that demographic parity difference between the source domain and the target domain can be bounded as:

$$
\Delta_{DP}^{\mathcal{T}} - \Delta_{DP}^{\mathcal{S}} \le L_2 \sum_f \gamma_f,
\tag{31}
$$

where $\gamma_f$ represents the sensitive group representation distance between the source domain and the target domain. The proof is at the end of A.5.

Thus to minimize $\Delta_{DP}^{\mathcal{T}} - \Delta_{DP}^{\mathcal{S}}$, we have to minimize $\sum_f \gamma_f$. FatraGNN utilizes the alignment module to minimize $\sum_{f,y} \epsilon_f^y$, then we show that $\sum_f \gamma_f$ can also be minimized at the same time. For better illustration, we draw Figure 9. In Figure 9, all solid circles represent sets of nodes with a certain sensitive attribute and label,

such as $\mathcal{S}_0^1$ denoting nodes in the source domain with the sensitive attribute 0 and label 1, and $\mathcal{T}_0^1$ denoting nodes in the target domain with the sensitive attribute 0 and label 1. We encircle nodes with the same sensitive attribute in the source domain and target domain with blue and orange circles, separately. For example, $\mathcal{S}_0$ represents nodes with sensitive attribute 0 in the source domain, and $\mathcal{T}_0$ represents nodes with sensitive attribute 0 in the target domain. Assuming that nodes with the same label have more similar representations, then the representation distance from $\mathcal{S}_0^0$ to $\mathcal{T}_0$ is the same as the distance from $\mathcal{S}_0^0$ to $\mathcal{T}_0^0$ ($\epsilon_0^0$), and the representation distance from $\mathcal{S}_0^1$ to $\mathcal{T}_0$ is the same as the distance from $\mathcal{S}_0^1$ to $\mathcal{T}_0^1$ ($\epsilon_0^1$). Thus the representation distance from $\mathcal{S}_0$ to $\mathcal{T}_0$ is denoted as $\gamma_0 = \max(\epsilon_0^0, \epsilon_0^1)$. Therefore, minimizing $\sum_{f,y} \epsilon_f^y$ can minimize $\sum_f \gamma_f$ at the same time.

Proof of Eq. (31):

The proof is similar to the proof of Theorem 3 in the Appendix. First, we characterize the difference of $\Delta_{DP}$ between the source domain and the target domain, denoted as $\Delta_{DP}^{\mathcal{S}} - \Delta_{DP}^{\mathcal{T}}$, by analyzing the accuracy difference between the source domain and the target domain of each sensitive group.

$$\Delta_{DP}^{\mathcal{T}} - \Delta_{DP}^{\mathcal{S}} = |\mathbb{E}_{\mathcal{T}_0} - \mathbb{E}_{\mathcal{T}_1}| - |\mathbb{E}_{\mathcal{S}_0} - \mathbb{E}_{\mathcal{S}_1}|$$
$$\leq ||\mathbb{E}_{\mathcal{T}_0} - \mathbb{E}_{\mathcal{T}_1}| - |\mathbb{E}_{\mathcal{S}_0} - \mathbb{E}_{\mathcal{S}_1}||$$
$$\leq |\mathbb{E}_{\mathcal{T}_0} - \mathbb{E}_{\mathcal{S}_0}| + |\mathbb{E}_{\mathcal{T}_1} - \mathbb{E}_{\mathcal{S}_1}|$$
$$= \sum_f |\mathbb{E}_{\mathcal{T}_f} - \mathbb{E}_{\mathcal{S}_f}|.$$

We then define sensitive group representation distance between the source domain and the target domain in Definition, and build a relationship between the representation distance and $|\mathbb{E}_{\mathcal{T}_f} - \mathbb{E}_{\mathcal{S}_f}|$.

$$\gamma_f = \max_{v_j \in \mathcal{T}_f} \min_{v_i \in \mathcal{S}_f} ||z_i - z_j||_2$$

To characterize the relationship between $\mathcal{T}_f$ and $\mathcal{S}_f$, we utilize a partition that can separate the nodes in $\mathcal{T}_f$ into different sets $R_f^{(i)}$. Every node $v_j$ in $R_f^{(i)}$ is near to a certain node $v_i \in \mathcal{S}_f$, satisfying $||z_i - z_j||_2 \leq \epsilon_f$. Obviously, $\mathcal{T}_f = \cup_{v_i \in \mathcal{S}_f} R_f^{(i)}$. As $\mathbb{E}_{\mathcal{S}_f^y} = \mathbb{E}_{v_i \in \mathcal{S}_f}(\omega(z_i)[y])$, $\mathbb{E}_{\mathcal{T}_f} = \mathbb{E}_{v_i \in \mathcal{T}_f}(\omega(z_i)[y])$, we have:

$$|\mathbb{E}_{\mathcal{T}_f} - \mathbb{E}_{\mathcal{S}_f}| = |\mathbb{E}_{i \in \mathcal{S}_f}(\hat{y}_i = y) - \mathbb{E}_{j \in \mathcal{T}_f}(\hat{y}_j = y)|$$
$$= |\mathbb{E}_{i \in \mathcal{S}_f}(\omega(z_i)[y]) - \mathbb{E}_{j \in \mathcal{T}_f}(\omega(z_j)[y])|$$
$$= \frac{1}{n_{\mathcal{S}_f}} \sum_{i \in \mathcal{S}_f} |\omega(z_i)[y] - \frac{1}{n_{R_f^{(i)}}} \sum_{j \in R_f^{(i)}} \omega(z_j)[y]|$$
$$= \frac{1}{n_{\mathcal{S}_f}} \sum_{i \in \mathcal{S}_f} \frac{1}{n_{R_f^{(i)}}} \sum_{j \in R_f^{(i)}} |\omega(z_i)[y] - \omega(z_j)[y]|.$$

As $\omega$ has $L_2$-Lipschitz continuity, $|\omega(z_i)[y] - \omega(z_j)[y]| = ||\omega(z_i) - \omega(z_j)||_\infty = \frac{1}{\sqrt{2}}||\omega(z_i) - \omega(z_j)||_2 \leq \frac{1}{\sqrt{2}}L_2||z_i - z_j|| \leq \frac{1}{\sqrt{2}}L_2\gamma_f$. Then

$$|\mathbb{E}_{\mathcal{T}_f} - \mathbb{E}_{\mathcal{S}_f}| \leq \frac{1}{\sqrt{2}}L_2\gamma_f \leq L_2\gamma_f,$$

we have:

$$\Delta_{DP}^{\mathcal{T}} - \Delta_{DP}^{\mathcal{S}} \leq \frac{1}{\sqrt{2}}L_2 \sum_f \gamma_f.$$

# B REPRODUCIBILITY INFORMATION
## B.1 Dataset Statistics

We use Bail-Bs, Credit-Cs, Pokecs, sync-B1s, and sync-B2s to evaluate our model as described in Section 5. The modularity-based community detection method we used to partition Bail [23] and Credit [46] is provided by Gephi. For each training graph in each dataset, we randomly choose 50% nodes for training and 25% nodes for validation. And we use all the nodes in all the testing graphs for testing. The statistics of Bail-Bs, Credit-Cs, Pokecs are shown in Table 3, Table 4, and Table 5.

sync-B1s contains a training graph B0 and thirty testing graphs with different $u$. sync-B2s contains a training graph B0 and thirty testing graphs with different $u$.

All the datasets can be found in https://anonymous.4open.science/r/FatraGNN-118F.

**Table 3: Bail-Bs statistics.**

| dataset | B0 | B1 | B2 | B3 | B4 |
|---|---|---|---|---|---|
| Nodes | 4686 | 2214 | 2395 | 1536 | 1193 |
| Edges | 153942 | 49124 | 88091 | 57838 | 30319 |
| Features | | | 18 | | |
| Sensitive attribute | | | Race | | |
| Label | | | Bail/no bail | | |

**Table 4: Credit-Cs statistics.**

| dataset | C0 | C1 | C2 | C3 | C4 |
|---|---|---|---|---|---|
| Nodes | 4184 | 2541 | 3796 | 2068 | 3420 |
| Edges | 45718 | 18949 | 28936 | 15314 | 26048 |
| Features | | | 13 | | |
| Sensitive attribute | | | Age | | |
| Label | | | High/low risk | | |

**Table 5: Pokecs statistics.**

| dataset | Pokec-z | Pokec-n |
|---|---|---|
| Nodes | 67,796 | 66,569 |
| Edges | 1,303,712 | 1,100,663 |
| Features | | 265 |
| Sensitive attribute | | Region |
| Label | | Working field |

## B.2 Baselines

The publicly available implementations of Baselines can be found at the following URLs:

- GCN: (MIT license) https://github.com/tkipf/gcn

- MLP: (MIT license) https://github.com/xmu-xiaoma666/External-Attention-pytorch/tree/master/model/mlp
- FairVGNN: (MIT license) https://github.com/YuWVandy/FairVGNN
- NIFTY: (MIT license) https://github.com/chirag126/nifty/tree/main
- EDITS: (MIT license) https://github.com/yushundong/EDITS
- EERM: (MIT license) https://github.com/qitianwu/GraphOOD-EERM

## B.3 Operating Environment

- Operating system: Linux version 3.10.0-693.el7.x86_64
- CPU information: Intel(R) Xeon(R) Silver 4210 CPU @ 2.20GHz
- GPU information: GeForce RTX 3090

## B.4 Inplementation Details

We use Pytorch to implement FatraGNN, for other baselines, we utilize the original codes from their authors and train the models in an end-to-end way. For all the models, we first train each model on the training graph with careful finetune, and then we test the performance of classification and fairness on the testing graphs. In each training iteration, we performed five training steps, including training the discriminator to discriminate the sensitive attribute, training the encoder to deceive the discriminator, training the encoder and classifier to ensure accuracy, training the feature generator to generate features, and training the encoder to obtain similar representations. The respective numbers of iterations for these steps were denoted as T1, T2, T3, T4, and T5. For the proposed FatraGNN, we search on epoch number T ranging from {400, 600}. For T1, T2, T3, T4, and T5, we test them ranging from {2, 5}, {8, 10, 12}, {5, 8}, {2, 5, 8}, {2, 5}, respectively. For the learning rate of the feature generator, discriminator, classifier, and encoder, we test them from {0.001, 0.005, 0.01, 0.05}. In order to get a bunch of modified graphs, we modify the structure of the training graph by randomly removing and adding the same number of edges. Every 10 epochs we choose one modified graph for training. For fair comparisons, we randomly run 5 times and report the average results for all methods. The code of FatraGNN can be found in https://anonymous.4open.science/r/FatraGNN-118F.

## B.5 Hyperparameter Setting

We implement FatraGNN in PyTorch, and list some important parameter values in our model in Table 6. We use Adam [33] as the optimizer for encoder $\kappa$, discriminator $\xi$, feature generator $\Gamma$, and classifier $\omega$ with different learning rates.

**Table 6: Hyperparameter Setting**

|  | T | T1 | T2 | T3 | T4 | T5 | $\Gamma$ lr | $\xi$ lr | $\omega$ lr | $\kappa$ lr |
|---|---|---|---|---|---|---|---|---|---|---|
| credit | 600 | 5 | 5 | 12 | 5 | 2 | 0.05 | 0.001 | 0.01 | 0.005 |
| bail | 400 | 5 | 5 | 12 | 8 | 5 | 0.05 | 0.001 | 0.005 | 0.005 |
| pokec | 400 | 2 | 5 | 10 | 2 | 5 | 0.05 | 0.001 | 0.01 | 0.01 |

## C ADDITIONAL RESULTS
### C.1 Results on Credit-Cs

Table 7 shows classification and fairness performance on Credit-Cs.

## C.2 Trade-off on Pokecs

The accuracy-fairness trade-off of fairness baselines and FatraGNN on Pokec is shown in Figure 11.

## C.3 Analysis of the Adversarial Module

We do experiments on Credit-Cs with five different sets of randomly chosen hyperparameters and random seeds and plot the loss versus epoch curve of the feature generator, the discriminator, and the classifier on Credit-Cs. As shown in Figure 10, the amplitude of loss gradually decreases as the number of epochs rises, indicating that it is actually not difficult to tune the parameters to achieve convergence and FatraGNN is not that susceptible to instability under varying random seeds.

## C.4 Analysis of the Representation Differences between the Training Graph and Testing Graphs

In order to plot the representations of the training graph and testing graphs of Bail-Bs using t-SNE [39], we first obtain the representation matrices of the training graph and testing graphs. Then we concatenate them into a single matrix and apply the same transformation to obtain a new two-dimensional matrix. Then, we separate these matrices corresponding to the original graphs. For each graph, we plot the node representations in one figure, with different colors indicating nodes belonging to different EO groups. As shown in Figure 13, the representations of nodes belonging to the same EO group on the training graph and testing graphs are close. This indicates that by minimizing the representation difference between the training graph and the testing graphs, the alignment module of our model can ensure the proximity of representations of the same EO group between the training graph and testing graphs, thereby guaranteeing fairness and accuracy on the testing graphs. We also propose an alignment score $\sum_{f,y} \mathbb{E}_{i \in \mathcal{S}_f^y, j \in \mathcal{T}_f^y} \frac{(z_i z_j)}{||z_i|| \cdot ||z_j||}$ to measure the alignment of the training graph and the testing graphs. The alignment scores between B0 and B1, B2, B3, and B4 are 3.89, 3.88, 3.81, and 3.86, respectively. However, when the generation module and alignment module are excluded, the alignment scores experience a decrease, measuring 3.85, 3.85, 3.73, and 3.83. This demonstrates the capacity of the generation and alignment modules to facilitate the encoder in learning similar representations for the same EO group between the training and testing graphs.

## C.5 Hyperparameter Study

We investigate the impact of epochs used in each training step. Figure 14 shows the overall score of FatraGNN on the Bail-Bs dataset. We can see that the performance benefits from an applicable selection of each epoch. Specifically, when T1 and T2 are set to small values, the adversarial module fails to guide the encoder in learning indistinguishable representations for sensitive groups. Similarly, a small T3 hinders the classifier's ability to effectively classify the nodes. In addition, a small T4 limits the generator's capacity to generate features that may lead to unfairness. Lastly, a small T5 prevents the alignment module from facilitating the encoder in

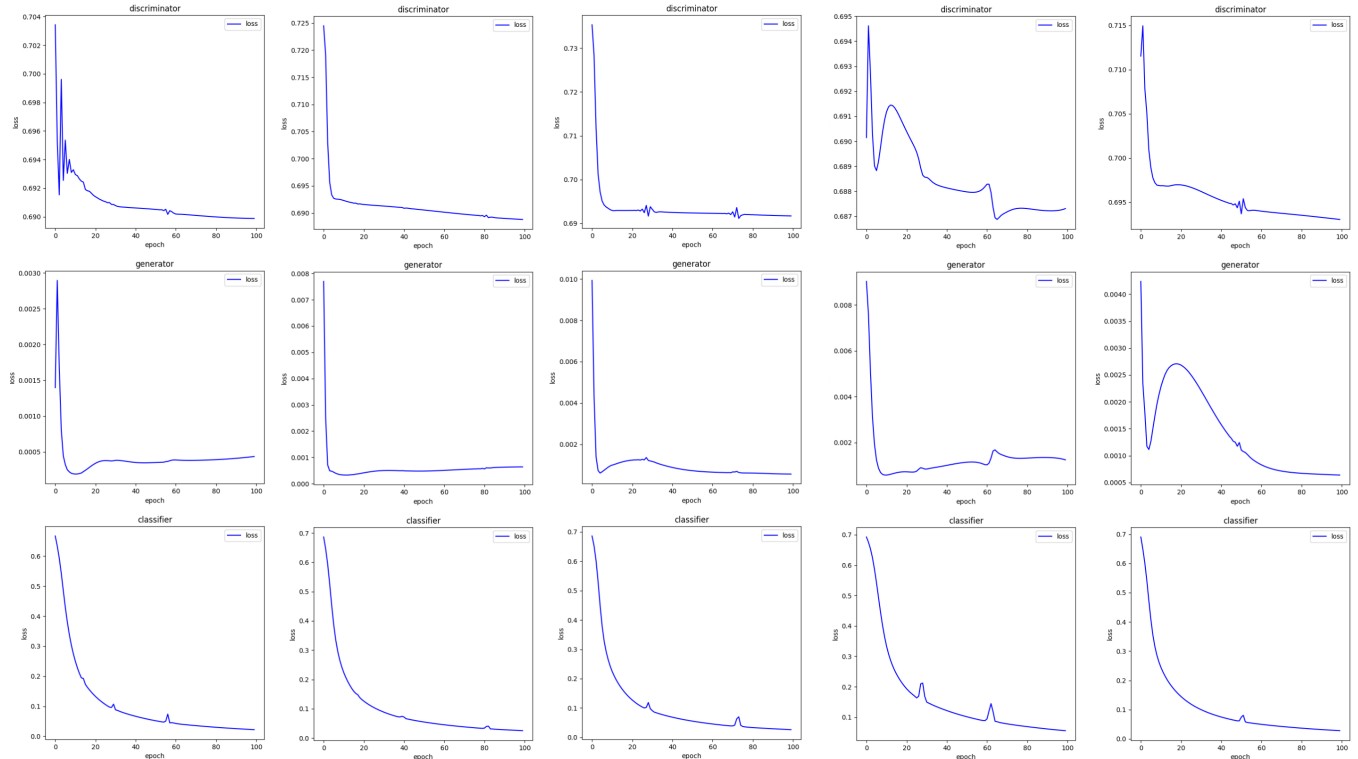

**Figure 10: Loss versus epoch curve of the feature generator, the discriminator, and the classifier on Credit with five different sets of randomly chosen hyperparameters and random seeds.**

**Table 7: Classification and fairness performance (%±σ) on Credit-Cs. ↑ denotes the larger, the better; ↓ denotes the opposite. Best ones are in bold.**

|  | metric | MLP | GCN | FairVGNN | NIFTY | EDITS | EERM | SWAD | SAGM | FatraGNN (ours) |
|---|---|---|---|---|---|---|---|---|---|---|
| C1 | ACC↑ | 75.69±5.64 | 76.69±2.48 | 77.45±0.31 | **77.51±0.03** | 77.06±0.03 | 77.04±0.09 | 76.98 ±0.35 | 77.16 ±0.45 | 77.31±0.1 |
|  | ROC-AUC↑ | 64.38±0.47 | 65.54±1.43 | 67.07±0.4 | **68.43±0.25** | 65.25±1.29 | 66.54±1.37 | 66.08±0.89 | 65.35±1.03 | 65.41±1.28 |
|  | $\Delta_{DP}$ ↓ | 5.2±14.92 | 7.4±6.69 | 1±0.63 | 4.34±0.03 | 2.43±0.03 | 5.46±0.38 | 4.37±0.35 | 5.21±0.76 | **0.5±0.21** |
|  | $\Delta_{EO}$ ↓ | 7.92±15.86 | 6.31±6 | 0.85±0.2 | 2.73±0.04 | 3.24±0.03 | 6.47±0.26 | 5.37±0.21 | 4.58±0.68 | **0.71±0.03** |
| C2 | ACC↑ | 72.05±2.6 | 75.42 ±0.44 | 75.49±1.7 | 74.44±0.47 | 77.07±0.22 | 76.16±0.91 | 75.64±0.98 | 76.38±0.91 | **77.12±0.28** |
|  | ROC-AUC↑ | 62.36±6.29 | 63.76±3.07 | 63.8±0.25 | 60.63±10.06 | 62.5±3.27 | 65.49±2.39 | 63.05±0.21 | 62.65±0.89 | **64.16±0.69** |
|  | $\Delta_{DP}$ ↓ | 8.14±4.39 | 8.74±3.6 | 3.54±0.42 | 3.54±1.6 | 2.98±0.01 | 4.22±1.38 | 3.48±0.91 | 5.34±0.67 | **1.64±1.06** |
|  | $\Delta_{EO}$ ↓ | 6.7±4.3 | 7.35±2.64 | 2.63±0.61 | 2.34±0.63 | 3.65±0.16 | 5.71±1.1 | 5.45±0.46 | 4.37±0.98 | **0.95±0.7** |
| C3 | ACC↑ | 68.15±0.16 | 70.31±1.79 | 71.49±0.56 | 70.11±0.04 | 70.89±0.96 | 71.43±1.24 | 70.34±0.29 | 70.83±0.65 | **71.81±0.39** |
|  | ROC-AUC↑ | 64.64±4.49 | 65.90±1.72 | **65.96±0.19** | 64.75±0.14 | 63.18±2.53 | 65.36±1.03 | 62.65±0.56 | 64.52±0.91 | 65.7±0.91 |
|  | $\Delta_{DP}$ ↓ | 8.7±0.12 | 9.46±10.06 | 3.05±1.76 | 3.54±0.07 | 3.22±0.45 | 5.63±9.43 | 5.36±0.76 | 5.12±1.36 | **0.25±0.2** |
|  | $\Delta_{EO}$ ↓ | 9.47±0.03 | 9.71±8.29 | 3.35±2.46 | 2.23±0.08 | 1.87±0.36 | 5.34±4.62 | 5.67±1.24 | 5.57±0.79 | **0.81±0.56** |
| C4 | ACC↑ | 68.26±3.09 | 70.89±5.38 | 71.74±0.45 | 71.84±6.36 | 71.28±0.2 | 71.35±4.28 | 71.56±0.87 | 71.22±0.29 | **72.15±0.42** |
|  | ROC-AUC↑ | 65.32±4.42 | 64.28±3.45 | 66.45±1.61 | 66.98±2.3 | 63.45±0.6 | 64.04±0.67 | 66.49±0.96 | 63.45±0.88 | **67.66±0.87** |
|  | $\Delta_{DP}$ ↓ | 7.46±12.08 | 6.13±4.08 | 3.46±0.05 | 7.84±9.64 | 3.42±0.47 | 4.35±0.85 | 4.85±0.79 | 5.46±1.18 | **0.61±0.08** |
|  | $\Delta_{EO}$ ↓ | 6.61±11.22 | 8.16±2.38 | 2.82±0.19 | 2.18±9.91 | 3.22±0 | 5.07±0.74 | 5.46±1.43 | 5.34±0.22 | **1.16±0.13** |
| | rank | 9 | 8 | 2 | 4 | 3 | 5 | 6 | 7 | **1** |

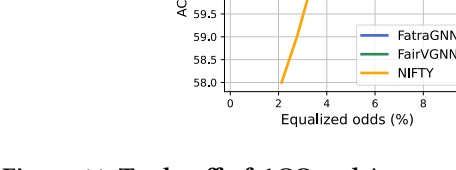

Figure 11: Trade-off of ACC and $\Delta_{EO}$ on the testing graph of Pokec. Upper-left corner (high accuracy, low $\Delta_{EO}$) is preferred.

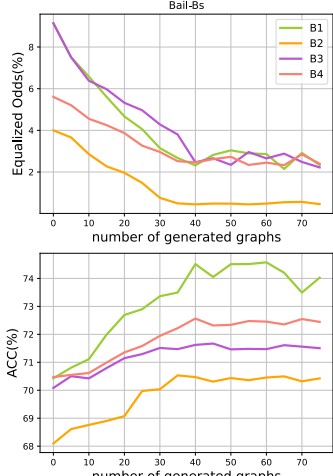

Figure 12: The hyperparameter sensitivity of FatraGNN with varying generated graph numbers.

learning similar representations for the initial graph and the generated graph, resulting in unfairness.

As we modify the graph structure to get graphs with different $u$ by adding and removing the edges of the graph, we also explore the impact of the edge modification ratio. As shown in Figure 15, we observe that FatraGNN achieves better performance when the ratio is around 0.5. Conversely, when the edge modification ratio is either too small or too large, FatraGNN's performance deteriorates. A small edge modification ratio results in little structural change in the generated graph, making it challenging for FatraGNN to adapt to testing graphs with different $u$. On the other hand, a large edge modification ratio leads to substantial changes in the graph structure and may disrupt the original information, making it difficult for FatraGNN to learn meaningful patterns.

Additionally, to investigate the optimal number of generated graphs, we keep other parameters fixed and vary the number of generated graphs utilized during training on Bail-Bs. As shown in Figure 12, the training process yields better results when the number of generated graphs reaches 40.

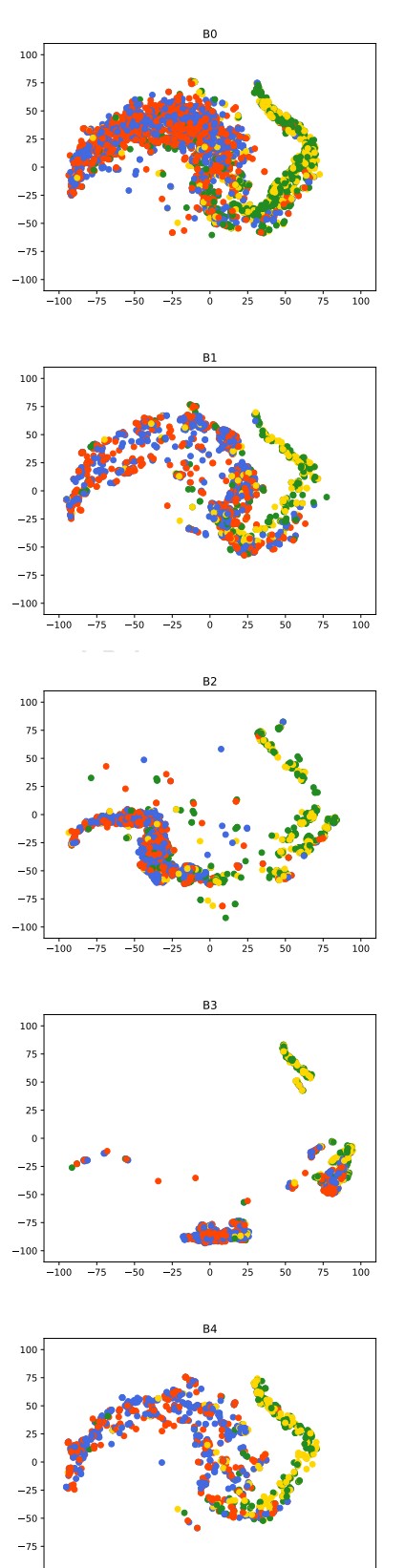

Figure 13: Visualization of the representations of the training graph and testing graphs.

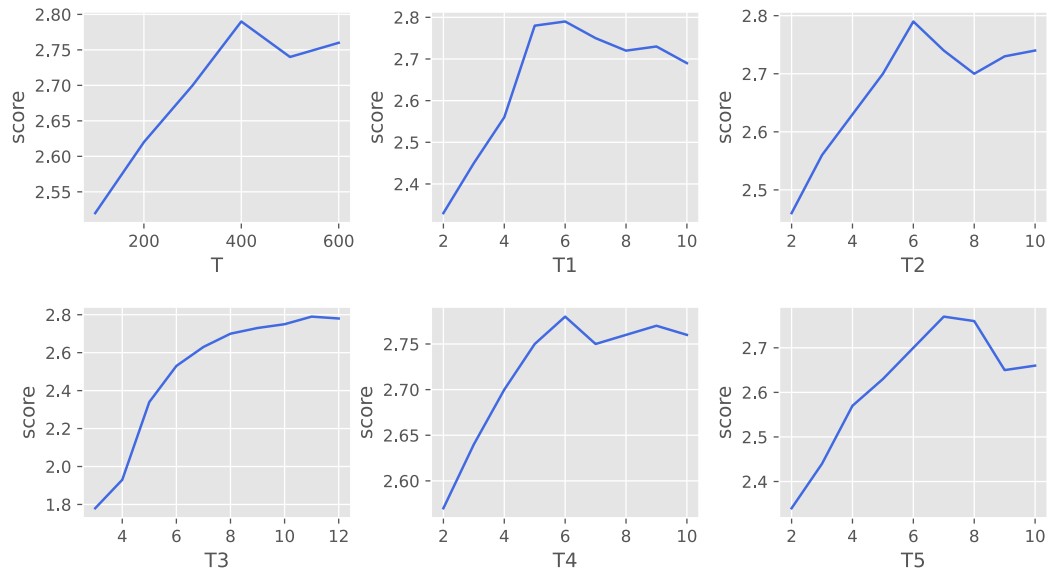

**Figure 14: The hyperparameter sensitivity of FatraGNN with varying epoch numbers of each training step.**

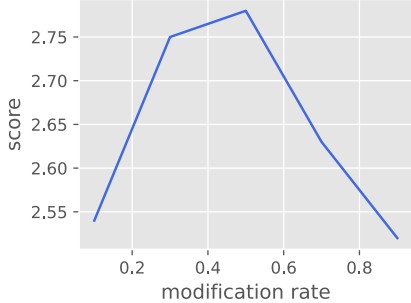

**Figure 15: The hyperparameter sensitivity of FatraGNN with varying modification rates.**

## D RELATED WORK

**OOD Generalization on Graphs** [26] proposes a novel out-of-distribution generalized graph neural network to solve the problem of generalization of GNNs under complex and heterogeneous distribution shifts. [44] focuses on out-of-distribution generalization for node-level problems and aims GNNs to minimize the mean and variance of risks from data with different distributions simulated by adversarial context generators. Although these works can release the problem of classification performance deterioration under distribution shifts, they may lead to unfairness.

