# OpenReview forum: "Graph Fairness Learning under Distribution Shifts"
_ACM.org/TheWebConf/2024/Conference — TheWebConf24_

### Official Review · Reviewer_pDJM · 2023-11-20

**Novelty:** 4
**Technical Quality:** 4

**Review:**

Overall, this is a very well-written paper with good contributions in the fair graph ML space. The authors provide a very thorough quantitative and qualitative evaluation, and present a straightforward training procedure.

Some strong points:

The evaluation is quite strong, against many different baseline models, and evaluating well on several measures. The authors provide a thorough ablation study.

Some weak points:

1. The authors don't sufficiently demonstrate that the datasets measure a shift in distribution, rather than a transductive learning setting. I dont feel that the measure u or 𝜂 captures the distribution shift fully.

2. The authors present datasets that are somewhat artificial to the distribution shift problem. The authors ought have used datasets where there is a real application use-case for learning under distribution shift. Instead, the authors primarily treat a graph partitioning as a distribution shift. Of the datasets, Bail and Credit seem like plausible fairness applications.

**Questions:**

How does the distribution shift setting differ from a subgraph or graph-level transductive learning, which has been a popular setting studied in many GNN models? Similarly, out-of-distribution GNN models (which the authors evaluate) aren't well-represented in related work. Fairness in GNN, Fairness under Distribution shift doesnt seem the interesting taxonomy split. Instead different distribution shifts in graphs (fairness seems a regularization on top of this).

**Reviewer Confidence:**

3: The reviewer is confident but not certain that the evaluation is correct

**Scope:**

3: The work is somewhat relevant to the Web and to the track, and is of narrow interest to a sub-community

---

### Official Review · Reviewer_9iUM · 2023-11-24

**Novelty:** 5
**Technical Quality:** 4

**Review:**

This paper focuses on graph fairness problem under shifted data distribution, which is a novel researching aspect.
As the training data and testing data sometimes does fall under different distribution, fairness under this problem is worth researching.

**Strength**
a) Clear organization: the paper has a clear logic and well-structured layout, making it easy for readers to follow.
b) Reasonable: they give concise explanation and detailed proof for theorems, which makes the following framework resonable and understandble.
c) Practical: They use realworld datasets for evaluation, which validates the model's effectiveness in practical scenarios.
d) Effectiveness: Provide extensive experiments with detailed explanation for the quantitative results, which show that their framework well-ensured the graph fairness on the unknown-distribution dataset.

**Weakness**
a) For the graph generation module, the motivation of the proposed two strategies for structure modification step is not well-mentioned.
b) For the evaluation metric s, lack detailed explanation.  Because these metrics are not in the same scale, simply adding them together without normalization is not that reasonable.
c) For out-of-distribution (OOD) GNN, only evaluates one model: EERM. There are other OOD GNN including using invariant learning, adversarial traning and self-supervised learning methods. Lake comparison among these models.
d) Focus on the binary classification task, which is too simple and not practical in most cases in real world. Not sure whether the proposed framework will still work under multi-classfication problem.
e) The analysis in ablation study is not sufficient. Not fully explained the graph results.

**Questions:**

a) For the graph generation module, why the two contradictory strategies can both simulate the shifted data distribution?
What about other strategies like randomly add edges and remove edges regardless the sensitive attribute?
When applying these random addition and removel, will some basic properties of the graph being modified or violated? For example, for drug discorvery, there are some chemical bonds that cannot be removed or modified.
b) For the metric s described in section five, why the simple addition instead of weighted sum or multiplex or other more complex combination?
c) In ablation study on Bail-Bs, for accuracy, why adversarial training makes the accuracy decrease in B2 and B3?
d) For the binary classification task, why use MLP classifier instead of others?

Typos:
Line 168: Equalized odds instead of "oods".

**Reviewer Confidence:**

2: The reviewer is willing to defend the evaluation, but it is likely that the reviewer did not understand parts of the paper

**Scope:**

4: The work is relevant to the Web and to the track, and is of broad interest to the community

---

### Official Review · Reviewer_snqt · 2023-11-25

**Novelty:** 4
**Technical Quality:** 4

**Review:**

### **Summary**

This paper studies graph fairness learning under distribution shift. The authors first provide theoretical analysis that distributional shift in training and test graphs could affect fairness, then propose a adversarial learning + graph generation module to achieve fairness under distribution shift. Experimental results on real-world and synthetic datasets demonstrate the effectiveness of the proposed method.

### **Strong Points**

S1. Graph fairness learning under distribution shift is rarely studied so this paper is studying important and novel research problem.

S2. The authors provide theoretical analysis on fairness under distribution shift

S3. Experimental results show that the proposed method can achieve fairness on graphs with different distribution.


### **Weak Points**

W1. Figure 1 brings no additional information. It should illustrate what issues would arise when deploying Fair GNN trained on one distribution on the unknown target graphs with different distributions. Right now, Figure 1 is just a pipeline to show fair GNN is deployed on unknown graphs.

W2. Motivation-wise, if the training graph and test graph are completely two **different** graphs (might be a pure inductive setting), what's the uniqueness of graph fairness learning? Isn't it just fairness under distribution shift (e.g., [3, 9, 15, 29, 35, 36] in the submission)? Please discuss the connections and differences to these works and compare them as baseline methods. Simplying saying this work is on graph data is not convincing because training graph and test graph are two different graph (rather than in the same graph as transductive setting). This is also evidential in your conceptual modeling of generation step where the distribution of generated (test) graph has nothing to do with training graph.

W3. Why is [34] in the submission degree-related fairness? There are many papers studying in this direction [1, 2, 3, 4, 5, 6, 7]. It is weird that none of them are cited, but citing one paper that seems not related to it directly.

W4. Group fairness has many other definitions other than equalized odds and demographic parity, e.g., equal opportunity, accuracy parity. Please correct your statement in lines 166 - 167.

W5. $y_i$ in lines 170 - 171 is never defined.

W6. In Eq. (2), why is it $\cup$ not $\cap$ if you want a node with $f_i = f$ **and** $y_i = y$? Additionally, both Eqs. (1) and (2) should have $\forall i$ or $\forall v_i$ to define the domain of $i$.

W7. Contents in Section 3.1 (the relationship between sensitive homophily and fairness) is also studied in existing works [8, 9]. The authors should discuss the connections and differences with these existing works.

W8. If you just consider the fairness as a utility metric, fairness discrepancy on training graphs and test graphs might have very setting to the error bound in transfer learning where we consider the discrepancy in loss function. Can the authors provide some discussions on that? E.g., what's the connections and differences? What's the superiority of your bound vs. existing unified generalization bound in tranfer learning (test error <= train error + distributional discrepancy + residual term)?

W9. If you generate graph by modifying topology and then generate features, the test graph still bears high similarity with training graphs. This is also evidential in your experimental settings, e.g., Pokec-n and Pokec-z are from the same social network; and Bails are from the same graph.

W10. The authors should compare graph transfer learning methods as baseline methods (in addition to vanilla GNN), to show that these transfer learning-enabled GNNs can still exhibit unfairness. Otherwise, if such GNNs can achieve low bias, what is the motivation of studying graph fairness learning under distribution shift.

W11. Another intuitive, important, but missing baseline method is to pick graph transfer learning method and simply add adversarial debiasing module as regularizer.


**References**

[1] Wu, Jun, Jingrui He, and Jiejun Xu. "Net: Degree-specific graph neural networks for node and graph classification." Proceedings of the 25th ACM SIGKDD International Conference on Knowledge Discovery & Data Mining. 2019.

[2] Tang, Xianfeng, et al. "Investigating and mitigating degree-related biases in graph convoltuional networks." Proceedings of the 29th ACM International Conference on Information & Knowledge Management. 2020.

[3] Liu, Zemin, Trung-Kien Nguyen, and Yuan Fang. "Tail-gnn: Tail-node graph neural networks." Proceedings of the 27th ACM SIGKDD Conference on Knowledge Discovery & Data Mining. 2021.

[4] Kang, Jian, et al. "Rawlsgcn: Towards rawlsian difference principle on graph convolutional network." Proceedings of the ACM Web Conference 2022. 2022.

[5] Wang, Ruijia, et al. "Uncovering the Structural Fairness in Graph Contrastive Learning." Advances in Neural Information Processing Systems 35 (2022): 32465-32473.

[6] Liu, Zemin, Trung-Kien Nguyen, and Yuan Fang. "On Generalized Degree Fairness in Graph Neural Networks." arXiv preprint arXiv:2302.03881 (2023).

[7] Yun, Sukwon, et al. "Lte4g: long-tail experts for graph neural networks." Proceedings of the 31st ACM International Conference on Information & Knowledge Management. 2022.

[8] Chen, April, et al. "Graph Learning with Localized Neighborhood Fairness." arXiv preprint arXiv:2212.12040 (2022).

[9] Lin, Xiao, et al. "BeMap: Balanced Message Passing for Fair Graph Neural Network." arXiv preprint arXiv:2306.04107 (2023).

**Questions:**

Please see weaknesses.

**Ethics Review Description:**

NA.

**Reviewer Confidence:**

4: The reviewer is certain that the evaluation is correct and very familiar with the relevant literature

**Scope:**

3: The work is somewhat relevant to the Web and to the track, and is of narrow interest to a sub-community

---

### Official Review · Reviewer_RASa · 2023-11-26

**Novelty:** 3
**Technical Quality:** 4

**Review:**

Summary
This paper aims to address the fairness issue of graph NNs when there exists a distribution shift between source and target. The goal is to minimize the disparity of EO in the target domain. The authors show the disparity of the target domain is upper bounded by that of the source domain and the distance between feature representation of the two domains, therefore, they propose a loss function to minimize the upperbound approximately. Experiments show the proposed method outperform some baselines.

Strength

- Theoretically, this work presents several theories for the upper and lower bounds for the proposed min max distance \eta_y. This work also shows results bounding EO and the difference of EO between training and test domains. They also find a case where the existing measure sensitive homophily does not work, when it is small, the aggregation features of two sensitive groups can still be distinguishable.

- Experiments show the proposed method outperforms a series of baselines on OOD generalization and in-distribution graph fairness.

Weakness

- For the theories: There is no discussion on the tightness of the bounds. In addition, these results seems not to be unique for graph data but also can be applied to iid data.

- For the methodology which generates graphs to mimic graphs from the target domain, I wonder if there is any guarantee on the generated labels. In addition, how can the authors guarantee that the graphs generated by adding/removing edges between nodes from the same/different sensitive groups can always fit the target graph distribution? Note that this is not trained jointly with the feature generator. Fig. 7 shows the generated graphs are different from the training, but they did not show they are similar to the target domain. As the last term in Eq.14 shows, the authors do not want the distribution of source and generated graphs to be too different, but they are quite different shown in Fig. 7, I am not sure whether it is good to have generated graphs like those in Fig. 7 when number of epochs is large.

- This work should compare their method with methods that work on fairness under distribution shift, even if they are not for graph data (e.g., [1]).

[1] https://arxiv.org/abs/2303.03300

**Questions:**

- L187 is it generation of Y or prediction of Y?

- In graph data, we often observe correlations between A and X, e.g., homophily, in this work, did the authors consider this when in their invariant joint distribution P(A,X|e)?

- It seems the max min distance (Eq.3) is not symmetric, does that mean we have to minimize it with both (v_a,v_b) and (v_b,v_a)?

- It would be better to have an iequality directly showing the relationship among \Delta_{EO}^{\mathcal{T}}, \Delta_{EO}^{\mathcal{S}} and the EO difference between source and target.

- It would be great to add experiments to show their method can really generate graphs that have similar distributions to the real target graphs.

**Reviewer Confidence:**

3: The reviewer is confident but not certain that the evaluation is correct

**Scope:**

4: The work is relevant to the Web and to the track, and is of broad interest to the community

---

### Official Review · Reviewer_C8r7 · 2023-11-28

**Novelty:** 4
**Technical Quality:** 4

**Review:**

Summary:

This paper delves into the domain of graph fairness learning, particularly under distribution shifts, a topic yet to be thoroughly explored from a theoretical standpoint. The authors theoretically prove that graph fairness learning is determined by two key factors, and further establish the relationship between fairness on the testing graph and two factors. Additionally, they introduce a model named FatraGNN designed to ensure fairness performance under distribution shifts on graphs. The experiments conducted on several datasets showcase the effectiveness of the proposed model.


Strengths:

1. Graph fairness learning is a crucial task in the graph learning community.
2. The authors contribute extensive theoretical analysis, laying a solid foundation for their proposed model.
3. The experiments appear to be comprehensive, effectively demonstrating the proposed model's effectiveness.


Weaknesses:

1. The paper should cite, discuss, and even compare with some recent studies, such as [1,2,3].

2. In Section 4.2, the authors propose generating a synthetic graph with unfairness under different distributions to assist in model adaptation from the training data to the testing data. However, it remains unclear how the connections between the synthetic graph and the testing graphs are ensured. In other words, is this generated synthetic graph significantly different from the testing graph, and why can this synthetic graph facilitate addressing fairness issues under distribution shifts? This raises questions about the relevance and applicability of the synthetic graph to the testing scenario. With the assistance of the synthetic graph, there is a concern that the learned model on the training graph might not be suitable for the testing graph.

3. The baselines used are somewhat outdated; recent approaches published in 2023 should be included for comparison.

[1] Towards Fair Graph Neural Networks via Graph Counterfactual. CIKM 2023 \
[2] On Generalized Degree Fairness in Graph Neural Networks. AAAI 2023 \
[3] Interpreting Unfairness in Graph Neural Networks via Training Node Attribution. AAAI 2023

**Questions:**

Please see the Review.

**Reviewer Confidence:**

3: The reviewer is confident but not certain that the evaluation is correct

**Scope:**

3: The work is somewhat relevant to the Web and to the track, and is of narrow interest to a sub-community

---

### Decision · Program_Chairs · 2024-01-22

**Decision:**

Accept

**Comment:**

The paper deals with what the authors claim (and I believe to be true) the new problem of out-of distribution fairness for graphs. This is a rather unexplored area that may have impact in several practical application domains.
 My reading of the results is positive and I also appreciate the mix of theoretical and simulation results. The reviewers also had a mostly positive opinion of the work. The major issue I had before the author's rebuttal was the comparison with other methods that are not tailor-made for graphs. The authors provided additional simulations that strengthened the case for their method.
 As a final comment, I strongly urge the authors to include the real-data comparative study involving general out-of-distribution fairness methods into their revision. Also, as pointed out by some reviewers, the revision should include more precise statements in Theorem 3.8 and more in-depth intuition/explanations regarding the theoretical results.